# Deuteration-enhanced neutron contrasts to probe amorphous domain sizes in organic photovoltaic bulk heterojunction films

Guilong Cai[1,2,11], Yuhao Li [1,3,4,11] ✉, Yuang Fu[1,11], Hua Yang[3,4], Le Mei[5], Zhaoyang Nie[6], Tengfei Li[7], Heng Liu[1], Yubin Ke[3,4], Xun-Li Wang [8,9], Jean-Luc Brédas [10], Man-Chung Tang [6], Xiankai Chen [5], Xiaowei Zhan [7] ✉ & Xinhui Lu [1] ✉

An organic photovoltaic bulk heterojunction comprises of a mixture of donor and acceptor materials, forming a semi-crystalline thin film with both crystalline and amorphous domains. Domain sizes critically impact the device performance; however, conventional X-ray scattering techniques cannot detect the contrast between donor and acceptor materials within the amorphous intermixing regions. In this study, we employ neutron scattering and targeted deuteration of acceptor materials to enhance the scattering contrast by nearly one order of magnitude. Remarkably, the PM6:deuterated Y6 system reveals a new length scale, indicating short-range aggregation of Y6 molecules in the amorphous intermixing regions. All-atom molecular dynamics simulations confirm that this short-range aggregation is an inherent morphological advantage of Y6 which effectively assists charge extraction and suppresses charge recombination as shown by capacitance spectroscopy. Our findings uncover the amorphous nanomorphology of organic photovoltaic thin films, providing crucial insights into the morphology-driven device performance.

In recent years, organic photovoltaics (OPVs) have made significant strides in power conversion efficiencies (PCEs) due to the development of non-fullerene acceptors (NFAs)[1–7]. With the synthesis of ITIC in 2015[1] and Y6 in 2019[6], the efficiency of single-junction devices has been progressively pushed over 19%[8–11]. A great deal of research has focused on understanding the intrinsic advantages of Y6-like molecules in exciton generation and diffusion, charge transfer and transport as well as in the recombination process[12–20]. These advantages arise directly from the unique chemical structure of Y6 molecules and their morphological features in the active layer. For instance, it has been reported that Y6-series molecules can form a two-dimensional (2D) network in the backbone plane due to the favourable end-group π-π stackings[21–23]. More recently, Brédas and co-workers used a combination of density functional theory calculations and molecular dynamics simulations to describe a local aggregation behavior of Y6 molecules in the amorphous donor and acceptor (D/A) intermixing region[24]. However, this phenomenon has not been observed experimentally, as there is a lack of electron density contrast detectable with conventional

[1]Department of Physics, The Chinese University of Hong Kong, Hong Kong, China. [2]Beijing Key Laboratory of Ionic Liquids Clean Process, Institute of Process Engineering, Chinese Academy of Sciences, Beijing, China. [3]Spallation Neutron Source Science Center, Dongguan 523803, China. [4]Institute of High Energy Physics, Chinese Academy of Sciences, Beijing 10049, China. [5]Department of Materials Science and Engineering, City University of Hong Kong, Hong Kong, China. [6]Institute of Materials Research, Tsinghua Shenzhen International Graduate School, Tsinghua University, Shenzhen 518055, China. [7]School of Materials Science and Engineering, Peking University, Beijing, China. [8]Department of Physics and Center for Neutron Scattering, City University of Hong Kong, Hong Kong, China. [9]Hong Kong Institute for Advanced Study, City University of Hong Kong, Hong Kong, China. [10]Department of Chemistry and Biochemistry, The University of Arizona, Tucson, Arizona 85721-0041, USA. [11]These authors contributed equally: Guilong Cai, Yuhao Li, Yuang Fu. ✉ e-mail: liyuhao@ihep.ac.cn; xwzhan@pku.edu.cn; xinhui.lu@cuhk.edu.hk

X-rays in the amorphous intermixing region. In other words, the fact that existing nanomorphology studies using X-ray scattering techniques[25,26] mainly rely on the electron density difference between crystalline and amorphous domains renders it incapable of probing characteristic lengths within the amorphous intermixing regions.

Neutron scattering has proven to provide a larger contrast than X-rays in the structural characterization of organic materials[27] because X-ray scattering cross-section is proportional to electron density, making it insensitive to light elements, while neutron interacts weakly with nuclei, resulting in an irregular variation in scattering length densities (SLDs) with atomic number. Small-angle neutron scattering (SANS), neutron reflectivity (NR) and quasi-elastic neutron scattering (QENS) have been employed to investigate the nanostructure, dynamic fluctuations of OPV active layers composed of fullerene acceptors since the dominant carbon components in fullerene derivatives provide sufficient SLD contrast relative to organic donor materials[28,29]. In the study by Dadmun et al., SANS was utilized for the first time to reveal the miscibility of PCBM in P3HT, the average PCBM domain size, and the interfacial area between PCBM and the P3HT-rich phases[30]. The results of SANS experiments conducted by Nedoma et al. indicated that the phase separation between PCBM and P3HT could be controlled by device fabrication conditions[31]. More following research also demonstrated the nanomorphology of fullerene-based organic solar cells by SANS[32–34]. In terms of the dynamic information, quasi-elastic neutron scattering (QENS) was utilized to monitor the motions of the side chains of P3HT polymers which were slowed down upon addition and further crystallization of the PCBM molecules[35,36]. In 1999, P. Müller-Buschbaum et al. developed grazing incidence small-angle neutron scattering (GISANS) to enhance the signal-to-noise ratio and scattering volume with the grazing incidence geometry[37]. Matthias et al. first applied GISANS in OPV studies and investigated the phase separation and molecular intermixing in the P3HT/PC$_{61}$BM bulk heterojunction[38]. Subsequently, this technique was applied by Guo et al. to study the impact of alcohol post-treatment on the inner phase structure of PTB7:PC$_{71}$BM blend films[39]. In 2018, W. Wang et al. used time-of-flight (TOF)-GISANS in the P3HT:PC$_{61}$BM bulk heterojunction thin film and quantitatively determined the molecular miscibility between P3HT and PC$_{61}$BM as well as the depth-dependent morphology changes induced by additives[28]. However, even with the aid of neutron scattering, it remains challenging to probe the nanomorphology of NFA-based blend films due to the similar hydrogen/carbon compositions of polymer donors and NFAs. The same issue also hinders the full characterization of ternary (and quaternary) blend films because the dopants used typically have similar chemical structures or the amount of dopants added is small. Therefore, although ternary strategies incorporating both fullerene[40] and NFAs (e.g. IDTBR[41] and Y6[42]) have proved effective in improving the efficiency and stability of OPVs, the exact microscopic origin remains controversial[43,44].

In this work, we aimed to overcome the challenge of low D/A contrast in OPV films by utilizing a combination of GISANS and targeted deuteration. This technique has been previously applied to study e.g., the structure of conducting polymers[45] and biological macromolecules[46], yet it has not been applied to probe the morphology of OPV active layers. We deuterated the prototypical high-performance NFA – Y6[6] and proved that it does not influence the morphological and optoelectronic properties of Y6 in both pure and blend films. Benefiting from the much larger coherent scattering cross-section of deuterium than hydrogen, a prominent increase in SLD and scattering intensity of NFAs was obtained, giving rise to nearly one order of magnitude enhancement of D/A contrast in the PM6:*d*-Y6 blend film. Intriguingly, a new scattering feature was detected for the first time by GISANS in the relatively large-*q* range in the PM6:*d*-Y6 blend film, indicating the presence of a short-range characteristic length in the active layer. This feature was not observed with X-rays,

suggesting that it does not originate from crystalline domains but rather from short-range *d*-Y6 aggregates embedded in the amorphous D:A intermixing domains. Similar aggregates were observed in the blend films of *d*-Y6 with other polymer donors as well as in the blend film of PM6 with deuterated Y7 (*d*-Y7), a chlorinated Y6 derivative[47], yet no such scattering feature was observed in the film of PM6 blended with another deuterated NFA - IDIC[48]. Through all-atom molecular dynamics simulations, we confirmed that this local aggregation behavior is an intrinsic morphological feature of Y6. The presence of the short-range aggregates within the intermixing domain was found to promote the formation of robust charge transport pathways that simultaneously suppress non-geminate recombination and improve charge extraction. This work demonstrates the power of GISANS combined with targeted deuteration labeling in studying the amorphous nanomorphology of organic photovoltaic thin films and the significance of this hitherto hidden aggregation morphological feature to the device performance.

## Results and Discussion
### Synthesis of deuterated nonfullerene acceptors and their optoelectronic properties

We followed the same synthesis route used for Y6[6] to synthesize deuterated Y6 (*d*-Y6), as shown in Fig. 1a. Deuterium was introduced through tributyl(6-undecylthieno[3,2-*b*]thiophen-2-yl)stannane (compound 1) and 2-(5,6-difluoro-3-oxo-2,3-dihydro-1H-inden-1-ylidene)malononitrile (2FIC) (Supplementary Fig. 1a and c). However, during the synthesis of deuterated 3-undecylthieno[3,2-b]thiophene-2-carboxylic acid (Compound S4 in Supplementary Fig. 1a), we observed unexpected hydrogen exchange for two deuterium atoms of the undecyl near the thiophene rings in the presence of hydrochloric acid. Nevertheless, the molecular deuterium substitution rate was only reduced marginally from 98% to 94%, which is considered negligible for the induced neutron contrast. The new compounds were fully characterized using $^1$H and $^{13}$C nuclear magnetic resonance (NMR) spectroscopy, mass spectrometry, and elemental analysis as shown in Fig. 1b, Supplementary Figs. 2-4, and Supplementary Note 1, respectively.

The Ultraviolet-visible (UV-vis) absorption spectra of Y6 and *d*-Y6 in the thin films were nearly identical, as depicted in Fig. 1c. Both acceptors exhibited two strong absorption peaks at 725 and 798 nm, indicating that deuteration does not affect the aggregation of Y6 molecules. Furthermore, we investigated the influence of deuteration on the molecular packing behavior of pure acceptor films using grazing-incidence wide-angle X-ray scattering (GIWAXS). The 2D GIWAXS patterns and the corresponding intensity profiles in the out-of-plane ($q_z$) and the in-plane ($q_r$) directions of pure *d*-Y6 and Y6 films are presented in Supplementary Fig. 10 and Fig. 1d, respectively. Both films displayed a preferential face-on orientation with the π-π stacking peak at $q_z = 1.77$ Å$^{-1}$ ($d = 3.55$ nm) and the backbone peaks at $q_r = 0.272/0.404$ Å$^{-1}$ ($d = 23.1/15.5$ nm), consistent with previous reports[6,22]. GIWAXS measurements were also performed on their blend films with the donor polymer PM6 (Supplementary Fig. 11). Both the PM6:Y6 and PM6:*d*-Y6 blend films exhibited a preferential face-on oriented molecular packing with π-π stacking peaks in the out-of-plane direction at $q_z = 1.76$ Å$^{-1}$ ($d = 3.57$ Å) and lamellar scattering peaks at $q_r = 0.302$ Å$^{-1}$ ($d = 20.8$ Å) (Supplementary Fig. 11). Additionally, the impact of deuteration on the electronic structure of Y6 was found negligible. We extracted the ionization energies (IEs) of *d*-Y6 and Y6 using ultraviolet photoelectron spectroscopy (UPS) (Supplementary Fig. 12a) and roughly estimated the corresponding electron affinities (EAs) by coupling them with the optical gaps (a procedure that neglects the exciton binding energies). The calculated IE/EA values of *d*-Y6 and Y6 are 5.64/4.29 and 5.59/4.2 eV, respectively. As a result, the device performance of OPV devices based on deuterated acceptors is almost identical to that of cells based on non-deuterated acceptors (Supplementary Fig. 12b).

In short, we demonstrated that deuteration does not notably affect the optoelectronic and morphological properties of the acceptors in both pure and blend films compared to the nondeuterated ones, making it a selective labeling technique feasible for probing the nanomorphology of organic photovoltaic bulk heterojunction (BHJ) active layers using GISANS.

## Scattering length densities of pure films under X-rays and neutrons

The SLDs of non-deuterated and deuterated acceptors under X-rays and neutrons were calculated from GISAXS and GISANS patterns, respectively. In the specular region of a typical grazing-incidence scattering pattern, we can identify the specular peak ($q_S = \frac{4\pi}{\lambda}\sin\alpha_i$) originating from the specular reflection as well as the Yoneda peak ($q_Y = \frac{4\pi}{\lambda}\sin\frac{(\alpha_i + \alpha_c)}{2}$) caused by surface enhancement[49]. Here, $\lambda$ represents the wavelength of the incident neutron beam, $\alpha_i$ is the incident angle, and $\alpha_c$ is the critical angle. By calibrating the incident angle based on the position of the specular peak and calculating the critical angle of the film via the formula for the Yoneda peak position, the SLD of the film can then be determined according to the formula $\rho = \pi\left(\frac{\alpha_c}{\lambda}\right)^2$[28].

First, GISAXS measurements were conducted for pure PM6, Y6, and $d$-Y6 films using an X-ray beam with a wavelength of 1.54 Å. Figure 2a, c and e show the measured 2D GISAXS patterns. The specular peaks of all films appeared at $q_z = 0.0498$ Å$^{-1}$ (marked as orange lines),

corresponding to an incident angle of 0.35°. The Yoneda peaks for pure PM6, Y6, and $d$-Y6 films were identified at $q_z = 0.0399$, 0.0397 and 0.0397 Å$^{-1}$, respectively (marked as blue lines), allowing us to estimate their X-ray SLD values as $1.76 \times 10^{-5}$, $1.74 \times 10^{-5}$ and $1.69 \times 10^{-5}$ Å$^{-2}$, respectively (Supplementary Table 1). It is worth noting that to avoid detector saturation, the exposure time measured without the vertical beamstop was significantly reduced. A more accurate Yoneda peak position can be identified in an off-axis vertical linecut of GISAXS patterns measured with the vertical beamstop and longer exposure time (Supplementary Fig. 13). As expected, the similar SLDs of these organic films confirm the lack of D/A contrast under X-rays.

Next, we carried out the time-of-flight (TOF)-GISANS measurements on the same PM6, Y6, and $d$-Y6 films. During each TOF-GISANS measurement, we collected neutron scattering signals from a broad wavelength range of 1.2-9.5 Å, from which single-wavelength images (with a bandwidth of 1.0 Å) were extracted for detailed analysis. Figure 2b, d and f show typical 2D GISANS patterns of pure PM6, Y6, and $d$-Y6 films using a neutron wavelength of 5 Å. The specular peak ($q_S$) and the Yoneda peak ($q_Y$) were also marked by the orange and blue lines, respectively. Since neutron flux used was much weaker than X-ray flux, a much longer exposure time was needed. Fortunately, the precision of the SLD derived could be improved by linearly fitting the calculated critical angles obtained at different wavelengths. The complete set of 2D TOF-GISANS patterns at different wavelengths is presented in Supplementary Fig. 14, and the extracted critical angles are plotted in

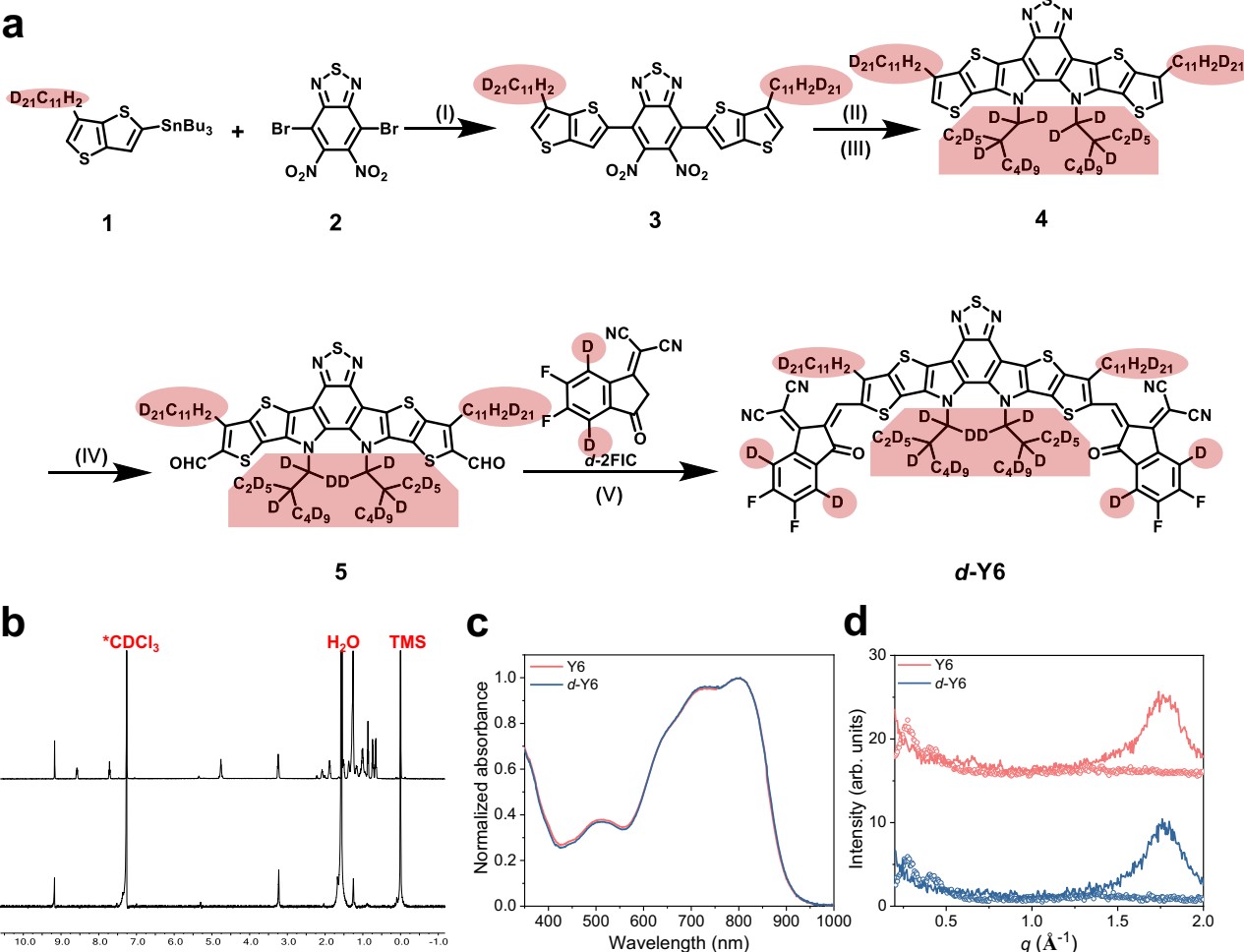

**Fig. 1 | Synthetic route and characterizations of pure films. a** Synthetic route of $d$-Y6. (I) Pd(PPh$_3$)$_2$Cl$_2$, Toluene, 80 °C; (II) triphenylphosphine, $o$-dichlorobenzene, 180 °C; (III) KI, K$_2$CO$_3$, DMF, 80 °C; (IV) POCl$_3$, DMF, 1,2-dichloroethane, reflux; (V) pyridine, chloroform, reflux. **b** $^1$H NMR spectra, **c** UV-vis absorption spectra, and **d** GIWAXS intensity profiles of Y6 and $d$-Y6 along the in-plane (dashed lines) and out-of-plane (solid lines) directions.

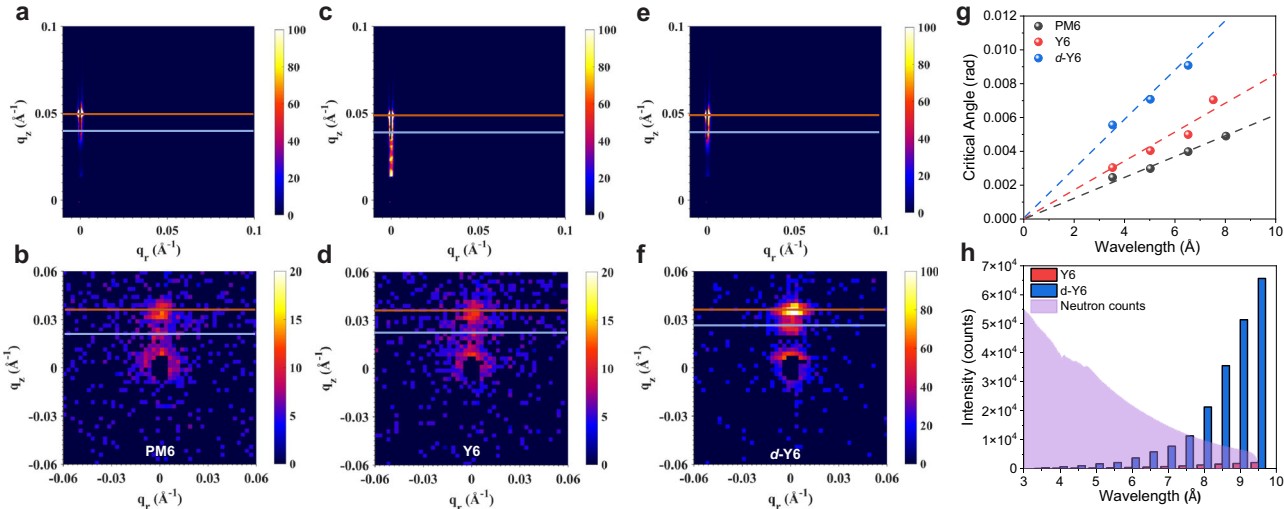

**Fig. 2 | GISAXS and TOF-GISANS measurements of pure films.** 2D GISAXS patterns of pure **a** PM6, **c** Y6, and **e** *d*-Y6 films with their corresponding 2D GISANS patterns measured with a neutron wavelength of 5.0 Å shown in **b**, **d**, and **f**. The positions of the specular and Yoneda peaks are marked by the orange and blue horizontal lines, respectively. **g** The extracted critical angles of PM6, Y6 and *d*-Y6 films as a function of neutron wavelengths (dots) with their best linear fits (dashed lines). **h** Neutron counts of the beam flux and the scattering of Y6 and *d*-Y6 at different neutron wavelengths.

Fig. 2g. Note that at certain wavelengths, the specular and Yoneda peaks are too close to each other to distinguish, and thus these wavelengths are excluded from the fitting. The SLDs of pure PM6, Y6 and *d*-Y6 films were fitted to be $(1.2 \pm 0.1) \times 10^{-6}$, $(2.3 \pm 0.1) \times 10^{-6}$ and $(6.4 \pm 0.4) \times 10^{-6}$ Å$^{-2}$, respectively (Supplementary Table 2). This suggests that the significantly increased SLD of *d*-Y6 compared to the non-deuterated Y6 could lead to nearly one order of magnitude increase in the D/A contrast, as the contrast scales proportionally to the square of the SLD differences ($I \propto \Delta\rho^2 = (\rho_D - \rho_A)^2$). Additionally, due to the much larger coherent scattering cross-section of deuterium than hydrogen, the neutron scattering counts of *d*-Y6 are orders of magnitude higher than those of the non-deuterated counterpart at all wavelengths (Fig. 2h), which could substantially improve the overall signal-to-noise ratio.

## Identification of amorphous domain size in blend films by GISANS

Equipped with the significantly enhanced D/A contrast, we then examined the nanomorphology of the PM6:*d*-Y6 blend film by GISANS and compared it with the results obtained from GISAXS. The films were prepared using chloroform (CF) containing 1-chloronaphthalene (CN, 0.5% v/v) as the solvent and were thermally annealed at 90 °C for 5 mins after casting. 2D GISAXS and GISANS patterns are presented in Fig. 3a and b, respectively. The scattering intensity and SLD of the PM6:*d*-Y6 blend film were higher than those of the non-deuterated counterpart under neutron beams (Supplementary Figs. 15 and 16), manifesting the effect of incorporating deuterated components.

Interestingly, the horizontal intensity profiles of GISAXS and GISANS were notably different, as shown in Fig. 3c and d, respectively. In the GISANS profile, the power-law intensity upturn that is commonly observed in the low-*q* region of the GISAXS profile does not appear. Instead, a new scattering feature emerges in the $q = 0.04-0.08$ Å$^{-1}$ range, indicating the presence of a hitherto hidden structure order that is exclusively accessible by neutron scattering. To aid a better understanding of our experimental results, schematics in Fig. 3e and f show the main features that give rise to scattering contrasts in GISAXS and GISANS, respectively. While X-rays primarily detect the contrast between crystalline and amorphous intermixing phases, neutrons can probe not only the Y6 crystalline phases, but also amorphous Y6 aggregates embedded within the intermixing phases due to the

targeted deuteration, which is anticipated to be the origin of the new scattering feature we detected. Based on this model, we fitted the intensity profile using Eq. 1:

$$I(q) = \frac{A_1}{[1 + (q\xi)^2]^2} + A_2 \left\langle P\left(q, R_{gc}\right) \right\rangle S_{frac}\left(q, R_{gc}, \eta, D\right) \\ + A_3 \left\langle P\left(q, R_{ga}\right) \right\rangle S_{hs}\left(q, R_{ga}, \Phi\right) + A_4 \tag{1}$$

where the first term represents the scattering contribution from the amorphous intermixing region (the Debye–Anderson–Brumberger, or DAB term[50]), the second term accounts for the contribution from the crystalline acceptor domains using the product of the spherical form factor and the fractal network structure factor[25], the third term models the amorphous acceptor domains with the spherical form factor and the hard-sphere structure factor, and the last term refers to a constant background. Details of the fitting model can be found in the Methods section.

Fitting the GISAXS profile (Fig. 3c) using Eq. 1 without the third term, we can extract the correlation length ($\xi$) of the amorphous intermixing phase and the size of the crystalline acceptor domain ($2R_{gc}$). On the other hand, the GISANS profiles can be well fitted by the second and third terms in Eq. 1 due to the highlighting of deuterated acceptors, giving rise to the crystalline acceptor domain sizes ($2R_{gc}$) and the amorphous acceptor domain sizes ($2R_{ga}$). As a result, the GISAXS-fitted $\xi$ and $2R_{gc}$ are 49.4 and 19.6 nm, respectively, while the GISANS-fitted $2R_{gc}$ and $2R_{ga}$ are 29.4 and 10.7 nm, respectively. Remarkably, a new structure feature of Y6 is thereby unveiled, providing the first experimental evidence to the simulation predictions of short-range aggregation of Y6 molecules in the amorphous intermixing phases[24]. We noted that the crystalline acceptor domain size extracted from X-ray scattering is smaller than that extracted from neutron scattering, which is possibly due to the miscounting of relatively amorphous crystalline domain boundaries by X-rays. In this regard, neutrons are more accurate in extracting the crystalline acceptor domain sizes[27]. Therefore, we will combine $2R_{gc}$ and $2R_{ga}$ extracted from GISANS and $\xi$ from GISAXS to compose a comprehensive physical picture of the nanomorphology in the active layer.

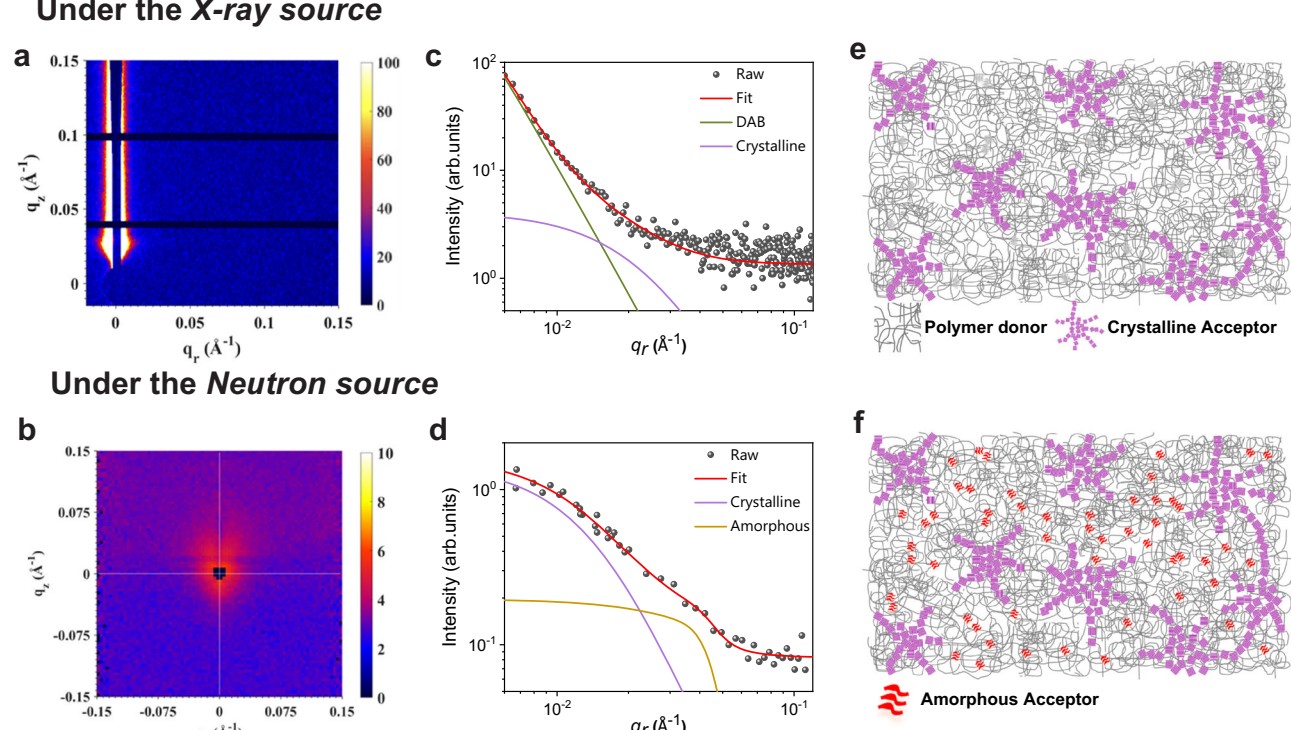

**Fig. 3 | GISAXS and TOF-GISANS measurements of blend films.** 2D **a** GISAXS and **b** TOF-GISANS patterns of PM6:*d*-Y6 with their horizontal linecuts (dots) with best fits (solid lines) shown in **c** and **d**, respectively. The terms DAB, Crystalline, and Amorphous stand for scattering contributions from D:A intermixing domains, Y6 crystalline domains, and amorphous Y6 aggregates, respectively. The schematics show the main features that give rise to scattering contrasts in **e** GISAXS and **f** GISANS measurements.

## The correlation between amorphous domain size and device performance

To correlate the extracted active layer nanomorphology with device performance, photovoltaic devices were fabricated conventional device with the structure of ITO /PEDOT:PSS/active layer/PNDIT-F3N/ Ag. The active layer is composed of a PM6:Y6 BHJ blend processed under three different conditions: as-cast with CF as the solvent (denoted as CF-ac), optimized conditions (90 °C thermal annealing and 0.5% v/v CN additives) with CF as the solvent (denoted as CF-opt), and optimized conditions (90 °C thermal annealing and 0.5% v/v CN additives) with chlorobenzene (CB) as the solvent (denoted as CB-opt). The *J-V* characteristics of OPV devices are shown in Fig. 4a, while the detailed device parameters and the fitted domain sizes of the corresponding deuterated active layers obtained from GISAXS and GISANS are summarized in Table 1. Under each processing condition, PM6:Y6 and PM6:*d*-Y6 devices show similar performance as shown in Supplementary Fig. 17 and Supplementary Table 3, suggesting that the GISANS results obtained from PM6:*d*-Y6 blend films are relevant for normal non-deuterated devices as well. The CF-opt device shows better device performance in terms of $J_{SC}$, $FF$, and $V_{OC}$ compared to the other two, which is consistent with a previous study[21]. In the next section, we will explain the difference in device performance in terms of active layer nanomorphology and charge carrier dynamics.

From CF-ac to CF-opt and CB-opt, the film formation time is gradually prolonged owing to the use of high-boiling solvent, solvent additives, and thermal annealing, resulting in distinct active layer morphology as demonstrated by both GISAXS and GISANS (Supplementary Figs. 18 and 19). Compared to CF-ac, CF-opt shows similar $\xi$ and $2R_{gc}$ but a noticeably increased $2R_{ga}$ from 7.8 to 10 nm, suggesting that the addition of high-boiling point solvent additive CN and mild thermal annealing mainly induce aggregation of Y6 molecules within the amorphous intermixing domains, which has a pronounced impact

on device performance as discussed later. Further prolonging the film formation time by replacing CF with CB results in the coarsening of Y6 crystalline domains with diameter increasing to around 58 nm as evidenced by the pronounced shoulder at low-*q* region (below 0.008 Å⁻¹) of the GISAXS intensity profile that can only be fitted via the hard-sphere model. Note that we used the GISAXS-fitted $2R_{gc}$ for the CB-opt film as the position of its main peak is beyond the *q*-range accessible by GISANS. The increase of $2R_{gc}$ is accompanied by the reduced $2R_{ga}$ back to 7.5 nm in the CB-opt film as these amorphous Y6 aggregates dissolve while the Y6 crystalline phases grow during the Ostwald Ripening process[50]. Tapping-mode atomic-force microscopy (AFM) was also applied to measure the surface topography of PM6:Y6 and PM6:*d*-Y6 films processed under the three aforementioned conditions as shown in Supplementary Fig. 20. The spherical agglomerates with sizes of around 50 nm in CB-opt is also visible from AFM, further supporting our GISAXS fitting results. However, AFM cannot identify the short-range Y6 aggregates observed from GISANS measurements as their sizes (5–10 nm) are below the resolution limit of the AFM tip we used (around 10 nm, see Methods).

To correlate the active-layer morphology with charge-carrier dynamics, capacitance spectroscopy was employed following the procedure developed by Brus et al.[51] from which the bias-dependent chemical capacitance and the charge carrier density can be derived (Supplementary Fig. 21a-c). The effective charge carrier mobility ($\mu_{eff}$) is plotted in the Fig. 4b as a function of the corrected applied bias ($V_{cor} = V_{app}$-$JR_s$, where $R_s$ is the series resistance derived from the dark *J-V* curve at +1.2 V). $\mu_{eff}$ can be linked to electron and hole mobilities via the equation[52] $\mu_{eff} = \frac{2\mu_e\mu_h}{\mu_e + \mu_h}$. Therefore, a higher $\mu_{eff}$ indicates a faster and more balanced charge transport. The CF-opt device shows the highest $\mu_{eff}$, followed by the CB-opt and CF-ac devices. Consistent results were also obtained via space-charge-limited-current (SCLC) measurements for single-carrier devices (Supplementary Fig. 22 and

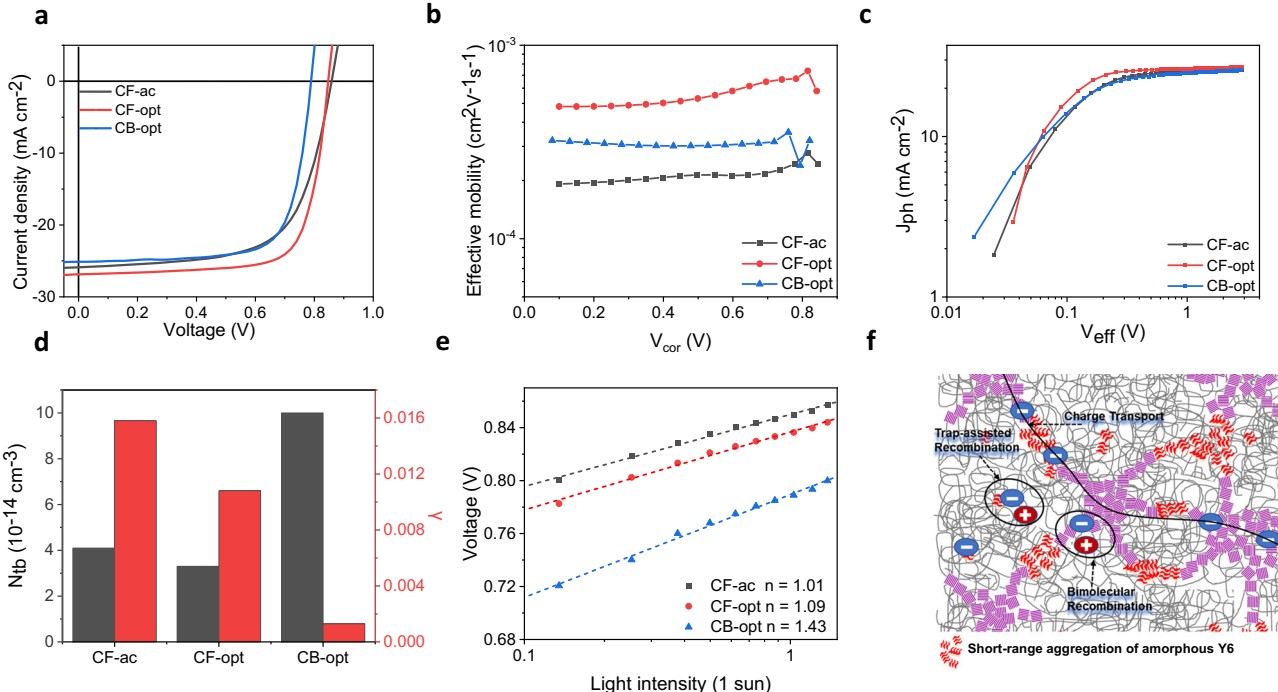

**Fig. 4 | Charge carrier extraction and recombination dynamics of devices. a** *J-V* curves of PM6:Y6 devices processed under three different conditions and **b** their effective mobilities ($\mu_{eff}$) derived as a function of corrected voltage ($V_{cor}$) from capacitance spectroscopy. **c** Photocurrent ($J_{ph}$) plotted against the effective bias ($V_{eff}$). **d** A direct comparison of the Langevin reduction factor ($\gamma$) and bulk trap state density ($N_{tb}$) between the three devices obtained from recombination current fittings. **e** Intensity-dependent $V_{OC}$ measurements. The ideality factors ($n$) obtained from fittings are highlighted. **f** Schematics showing different recombination mechanisms in PM6:Y6 devices.

### Table 1 | A summary of fitted domain sizes and device characteristics of PM6:Y6 systems

| Samples [a] | $\xi$ (nm) | $2R_{gc}$ (nm) | $2R_{ga}$ (nm) | $V_{OC}$ [b] (V) | $J_{SC}$ [b] (mA cm⁻²) | FF [b] (%) | PCE [b] (%) |
|---|---|---|---|---|---|---|---|
| CF-ac | 45.6 | 33.1 | 7.8 | 0.865 (0.861 ± 0.003) | 26.9 (26.7 ± 0.2) | 69.7 (69.0 ± 0.5) | 16.3 (15.9 ± 0.3) |
| CF-opt | 49.4 | 29.4 | 10.7 | 0.853 (0.850 ± 0.002) | 28.0 (27.7 ± 0.5) | 74.4 (74.1 ± 0.4) | 17.8 (17.4 ± 0.3) |
| CB-opt | N/A | 58.1[c] | 7.5 | 0.806 (0.798 ± 0.006) | 25.5 (24.7 ± 0.6) | 72.2 (71.5 ± 0.5) | 14.8 (14.1 ± 0.5) |

[a]D/A = 1/1.2 (w/w). [b]Average values with their standard deviations (in parentheses) are obtained from 15 independent devices. [c]The $2R_{gc}$ of the CB-opt film is obtained from the GISAXS fitting.

Supplementary Table 4). The use of CN and thermal annealing have been previously demonstrated to enhance the crystal coherence length along the π−π stacking direction[53] and to assist the formation of 2-D charge-transport networks[21]. Here, we propose that the presence of larger-size amorphous Y6 stackings in the intermixing phases is another key reason behind the improved carrier mobility of the CF-opt device as it will assist the formation of well-percolated conduction pathways in intermixing phases[24,54–56].

To elucidate the impact of active layer nanomorphology on non-geminate recombination, the recombination current $J_{rec}$, which is the difference between the saturated photocurrent ($J_{ph}$, see Fig. 4c) obtained at −2 V and $J_{ph}$ obtained under different bias conditions, is fitted with a linear combination of three different recombination currents[57]: bimolecular, bulk trap-assisted, and surface trap-assisted recombination as shown in Eqs. 2–4. The fitting results are shown in Supplementary Fig. 21d-f.

$$J_{bm} = qLk_{bm}n^2 = \frac{2q^2L}{\varepsilon_0\varepsilon_r}\gamma\mu_{eff}n^2 \qquad (2)$$

$$J_{bulk} = qLk_{tb}n = \frac{q^2L}{\varepsilon_0\varepsilon_r}\mu_{eff}N_{tb}n \qquad (3)$$

$$J_{surf} = qLk_{ts}(V_{cor})n = \frac{q^2}{\varepsilon_0\varepsilon_{rexp}}\frac{\mu_{eff}N_{ts}n}{\left\{\frac{q(V_{bi}-V_{cor})}{kT}\right\}} \qquad (4)$$

Here, $V_{bi}$ represents the built-in field, $q$ is the elementary charge, and $L$ is the active layer thickness. The three fitting parameters $\gamma$, $N_{tb}$, and $N_{ts}$ are the Langevin reduction factor for bimolecular recombination, bulk trap state density, and surface trap state density, respectively. As summarized in Fig. 4d, the values of $\gamma$ are lower in CF-opt (0.011) than in CF-ac (0.016), consistent with the previous study[53]. Considering the similar $\xi$ and $2R_{gc}$ in those two systems, the reduced $\gamma$ in the CF-opt should mainly arise from the larger amorphous Y6 aggregates that reduces the interfacial area for bimolecular recombination. On the other hand, CB-opt has an extremely low $\gamma$ of 0.001 due to its strongly phase-separated structure as evident from the GISAXS result. Apart from bimolecular recombination, the CF-opt device also has a lower $N_{tb}$ of 3.3 × 10¹⁴ cm⁻³ compared to CF-ac (4.1 × 10¹⁴ cm⁻³)

although this is not reflected from the intensity-dependent $V_{OC}$ measurements (Fig. 4e) due to its concomitantly suppressed bimolecular recombination. It is anticipated that the presence of larger amorphous Y6 aggregates (10.5 v.s. 7.8 nm) may help the formation of well-percolated conduction pathway and avoid the presence of isolated domains that may act as electron traps[58]. It is notable that the CB-opt device has the highest $N_{tb}$, $10^{15}$ cm$^{-3}$, that is in line with a much larger ideality factor of 1.43 compared to those of CF-ac (1.01) and CF-opt devices (1.09) (Fig. 4e). We attribute this to the greater phase separation of PM6 and Y6 in the CB-opt condition which results in the dominance of monomolecular recombination[59]. Overall, the large amorphous Y6 aggregates could form robust charge transport networks in the amorphous intermixing domains, which simultaneously improved charge extraction and suppressed charge recombination. As a result, the CF-opt device has the longest carrier lifetime ($\tau$) and the fastest charge extraction time ($\tau_{ext}$) as shown in Supplementary Fig. 21 h, resulting in the lowest non-geminate recombination loss as quantified by the competition factor ($\theta = \frac{\tau_{ext}}{\tau}$) among the three devices (Supplementary Fig. 21i), accounting for its superior FF and $J_{SC}$ values.

### Intrinsic short-range aggregation of Y6

To find out whether this short-range aggregation behavior is unique to Y-series NFAs, we performed deuteration substitution on Y7, a chlorinated Y6 derivative[47], and IDIC, another high-performance NFA modified from ITIC[48]. The synthesis of $d$-Y7 follows the same route as $d$-Y6 while the synthesis route of $d$-IDIC can be found in Supplementary Fig. 1d. The corresponding $^1$H and $^{13}$CNMR spectra are shown in Supplementary Fig. 5-8, while the mass spectroscopy and elemental analysis for $d$-IDIC can be found in Supplementary Fig. 9 and the Supplementary Note 2, respectively. Upon deuteration, the neutron SLDs increase from $1.77 \times 10^{-6}$ to $5.37 \times 10^{-6}$ Å$^{-2}$ and $1.85 \times 10^{-6}$ to $1.09 \times 10^{-5}$ Å$^{-2}$ for Y7 and IDIC, respectively. However, the scattering feature related to short-range aggregates was only detected in the blend film of PM6:$d$-Y7 but not in PM6:$d$-IDIC, as shown in Supplementary Fig. 23, 24 and Supplementary Table 5. This suggests that the inherent molecular structure of Y-series molecules is responsible for the formation of short-range aggregates, which is in line with the distinct local morphology predicted via molecular-dynamics simulations for pure films of Y6 compared to ITIC derivatives[24].

To investigate the impact of different donor polymers on nanomorphology, we mixed $d$-Y6 with three other typical donors P3HT, PTB7-Th, and J71. TOF-GISANS measurements were performed on these polymer:$d$-Y6 blend films, as shown in Supplementary Fig. 25a-c. The horizontal linecuts with their best fits and the corresponding contributions from two sets of parameters are illustrated in Supplementary Fig. 25d-f and Supplementary Table 5. The scattering feature due to the amorphous Y6 aggregates was well observed in all polymer:$d$-Y6 blend films, implying that the Y6 readily forms short-range aggregates in the intermixing phase regardless of the polymer donor species. The amorphous acceptor domain sizes in the blend films of Y6 with P3HT, PTB7-Th and J71 were calculated to be slightly smaller than that in the blend film with PM6, consistent with the reported stronger recombination[60].

To gain further insights into the formation mechanism of amorphous Y6 aggregates, all-atom molecular dynamics (AAMD) simulations were performed. The detailed methodology can be found in the Method section while the initial and final (optimized) configurations of all simulated systems can be found in Supplementary Data 1-6. Since deuteration has little impact on the morphology in the intermixing phase, in our AAMD simulations, deuterium atoms were replaced by hydrogen atoms. The simulation results show that the packing of Y6 molecules is similar in all three different blends, with typical packing modes involving terminal-terminal interactions (TT), core/terminal interactions (CT-CT), and core/core and terminal/terminal interactions (CC-TT) (Fig. 5a-c). The Y6 molecule was then decomposed into A, D, A'

fragments, and the radial distribution functions, $g(r)$, describing the packing between these molecular fragments (i.e., A-A, A-D, A-A', D-D, D-A', and A'-A') were collected from the three simulated blends. As shown in Fig. 5d-i, the $g(r)$ data indicated a strong intensity of the first peak below 4 Å for all the inter-fragment packings in the three blends. Thus, the simulation results confirm that, instead of being solubilized between amorphous PM6 fragments, Y6 molecules do aggregate through various short-range π-π stackings even in the amorphous intermixing domains, which is fully consistent with our experimental results. Since this unique feature has been observed in blend films of both Y6 and Y7, it likely arises from their excellent backbone planarity compared to ITIC-based NFAs. It has been shown that the beta side chains attached to the outer core groups of Y6 molecules, regardless of their chain lengths and shapes, can induce steric hindrance effects to suppress dihedral angles between core and end groups[61,62]. Therefore, we anticipate that our conclusion should be generally applicable to Y-series NFAs with side chains attached to the same position.

In summary, we demonstrated that deuteration has minimal effects on the morphology of NFAs, while providing enhanced D/A contrast in GISANS measurements, which allows much better detection of the amorphous nanomorphology in OPV active layers. Consequently, the nanomorphology of the PM6:$d$-Y6 blend film was examined using GISANS, uncovering the presence of amorphous Y6 aggregates embedded within the amorphous intermixing regions. By analyzing the scattering intensity profiles, the sizes of the amorphous acceptor domains under different fabrication conditions were determined and correlated with device performance. Our findings indicate that larger amorphous acceptor aggregates within the amorphous intermixing region facilitate the formation of interconnected charge transport pathways, enhancing charge extraction and suppressing non-geminate recombination. Notably, both experimental and theoretical results emphasized that the short-range aggregation of Y6 molecules in the intermixing phase of polymer/Y6 blends is an intrinsic feature of Y6 and remains little influenced by the specific type of polymer used. This unique morphological advantage of Y6 is particularly noteworthy considering the semi-crystalline nature of most organic bulk heterojunction systems, making the short-range aggregation a key contributor to its high photovoltaic performance. We anticipate that this targeted deuteration highlighting scheme will inspire further explorations into complex morphology systems, such as ternary and quaternary systems. In addition, further targeted deuteration at specific positions of molecules provides the opportunity to investigate local interaction mechanisms between different functional groups of organic donor and acceptor molecules.

## Methods

### Materials

Unless stated otherwise, all the chemical reagents and solvents used were obtained commercially without further purification. Chloroform (99.5%), 1,8-diiodooctane (DIO) (97.0%) and 1-chloronaphthalene (CN) (97.0%) were purchased from Sigma-Aldrich. PM6 ($M_n$ = 20.2 kg mol$^{-1}$, $M_w/M_n$ = 2.0), Y6, PNDIT-F3N and compounds S10 and 2 were purchased from OptiFocus Ltd. 1-Bromohexane-D13 was purchased from J&K Chemical Inc. 1-Bromo-2-ethylhexane-D17 and dodecanoic-D23 acid were purchased from CFW Laboratories, Inc. IDIC was synthesized by our previous procedure[48].

### Characterizations of deuterated materials

The $^1$H and $^{13}$C nuclear magnetic resonance (NMR) spectra were measured by Bruker AVANCE 400 and 500 MHz spectrometers. Mass spectra were performed by a Bruker Daltonics Biflex III MALDI-TOF Analyzer in the Matrix-assisted laser desorption/ionization (MALDI) mode. Elemental analyses were performed by a FlashSmart Elemental Analyzer. The ultraviolet-visible light (UV-vis) absorption spectra were

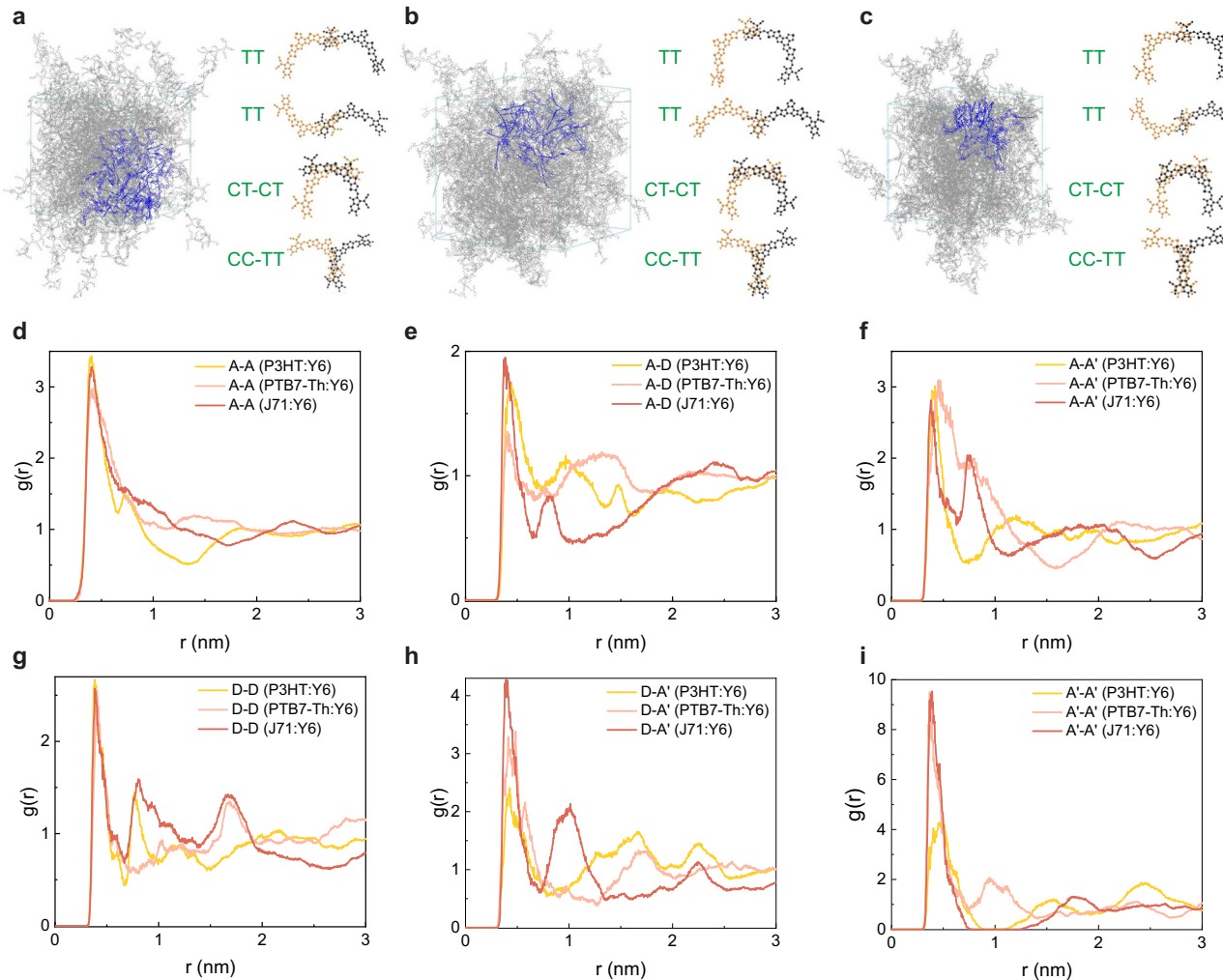

**Fig. 5 | All-atom molecular dynamics (AAMD) simulations of the Donor:Y6 blend systems. a–c** Illustration of stacking patterns in P3HT:Y6, PTB7-Th:Y6, and J71:Y6, respectively. Three typical dimers with terminal-terminal interactions (TT), core/terminal interactions (CT-CT) and core/core and terminal/terminal interactions (CC-TT) are extracted from the blue areas. **d–i** Radial distribution function ($g(r)$) data describing the packing between the Y6 molecular fragments (*i.e.*, A-A, A-D, A-A', D-D, D-A', and A'-A', respectively) for P3HT:Y6, PTB7-Th:Y6, and J71:Y6.

measured using the JASCO-570 spectrophotometer (JASCO. Inc., Japan) for thin films deposited on quartz substrates.

## Ultraviolet Photoelectron Spectroscopy (UPS)
UPS spectra were obtained from an AXIS ULTRA DLD (Kratos) with He I (21.22 eV) excitation lines and a sample bias of −9 V under a vacuum of $3.0 \times 10^{-8}$ Torr. Radiation damage from the light source on the organic films was carefully examined and no damage was detected. All measurements were carried out in the dark.

## Device fabrication
All the fabricated devices were based on a conventional sandwich structure, patterned ITO glass/PEDOT:PSS/active layer/ PNDIT-F3N/Ag. The ITO substrates were first scrubbed by detergent and then sonicated consecutively with deionized water, acetone, and isopropanol, and dried in an oven. The glass substrates were treated by UV-ozone for 20 min before use. PEDOT:PSS (Heraeus Clevios P VP AI 4083) was spin-cast onto the ITO substrates at 4000 rpm for 30 s, and then dried at 120 °C for 20 min in air. The donor/acceptor blends (1/1.2 weight ratio, total concentration is 15 mg mL⁻¹) were dissolved in various solvents (chloroform or chlorobenzene, with or without CN as the additive) and stirred overnight at room temperature in a nitrogen-filled glove box. The blend solution was spin-cast at 3000 rpm for 30 s.

Active layers were annealed on a 90 °C hotplate for 5 minutes after casting. A thin PNDIT-F3N layer ( ~ 5 nm) was coated on the active layer, followed by the deposition of Ag electrode (100 nm). The device area was determined by the overlap between the top and bottom electrodes, which is 0.04 cm⁻² as confirmed by the optical microscope. The solar-cell performance were tested within a nitrogen-filled glovebox without encapsulation. An Air Mass 1.5 Global (AM 1.5 G) solar simulator (SS-F5-3A, Enlitech) was used with an irradiation intensity of 100 mW cm⁻², which was measured by a calibrated silicon solar cell (SRC2020, Enlitech). The *J*–*V* curves were measured along the forward scan direction from −0.2 to 1.2 V, with a scan step of 20 mV and a dwell time of 10 ms, using a Keithley 2400 Source Measure Unit.

## GIWAXS and GISAXS measurements
GIWAXS and GISAXS measurements were carried out with a Xeuss 2.0 SAXS/WAXS laboratory beamline using a Cu X-ray source (8.05 keV, 1.54 Å) and a Pilatus3R 300 K detector. The incidence angle was 0.2°. All the samples were spin-coated on the silicon substrates with the same solution recipes as for the device active layers.

## TOF-GISANS measurements
TOF-GISANS was conducted at BL-01 (SANS) beamline at the China Spallation Neutron Source (CSNS). The TOF mode allows both

specular and off-specular scattering experiments, in which neutrons with a broad range of wavelengths are used simultaneously and recorded as a function of their respective times of flight. The depth information of sample was investigated by controlling the incident angle at three different values, namely $\alpha_i = 0.3°$, $0.6°$ and $0.9°$. The sample-to-detector distance was 4 m, and the chosen wavelength range was 1.2 Å -9.5 Å. All samples were measured for 15 h to obtain sufficient statistics.

The TOF-GISANS profiles are fitted with a universal model as shown in Eqs. 5 and 6:

$$I(q) = \frac{A_0}{[1 + (q\xi)^2]^2} + A_1 \langle P(q,R) \rangle S(q,R,\eta,D) + A_2 \langle P(q,R) \rangle S_{hs}(q,R,\Phi) + B \tag{5}$$

$$S(q) = 1 + \frac{Sin[(D-1)]\tan^{-1}(q\eta)}{(qR)^D} \frac{D\Gamma(D-1)}{[1 + \frac{1}{(q\eta)^2}]^{(D-1)/2}} \tag{6}$$

where the first term is the Debye−Anderson−Brumberger (DAB) term, q is the scattering wave vector, $A_0$ is an independent fitting parameter, and $\xi$ is the average correlation length of the amorphous domain. The second term represents the contribution from crystalline Y6 domains. Here, $P(q, R)$ is the form factor of the clusters. $S(q, R, \eta, D)$ is the structure factor of crystalline Y6 domains modeled as fractal-like networks. The third term represents the contribution from amorphous Y6 aggregates. Here, $S_{hs}(q,R,\Phi)$ is the structure factor calculated under Percus−Yevick Approximation which models amorphous Y6 aggregates as hard spheres.

## AFM measurements

JPK NanoWizard NanoOptics atomic force microscope (AFM) with a Tap300Al-G tip (40 N/m) was used for topography characterization of blend films. Measurements were carried out using non-contact mode with a piezoelectric tip oscillating at a fixed frequency (300 kHz) above the sample surface. For each measurement, tip was scanned over 256 pixels across a 2 μm range at a rate of 1 Hz. A 2×2 μm image was obtained for each sample. The root-mean-square (RMS) height fluctuations were obtained using the JPKSPM Data Processing software package.

## All-atom molecular dynamics simulation

The all-atom molecular dynamics simulations were carried out with the GROMACS 2021.2 package[63]. The 'optimized potentials for liquid simulations-all atom (OPLS-AA)' force field[64,65], which is a well-established force field for extended organic π-conjugated molecules, were tuned for the Y6 molecule as well as donor P3HT, PTB7-Th and J71 polymers. The structures of the P3HT, PTB7-Th, and J71 monomer and Y6 are shown in Supplementary Fig. 26. Each chain was built by 40, 8, and 8 repeat units for the P3HT, PTB7-Th, and J71 polymers, respectively. Partial charges for all the molecules were obtained by the restrained electrostatic potential (RESP) fitting method. All the Density Functional Theory (DFT) calculations were performed using Gaussian 16[66]. The initial models were built by randomly placing donor polymers chains and Y6 molecules in cubic cells with an extremely low density of ~0.1 g/cm³. 300 Y6 molecules were first inserted into each simulation box and then a total of 55 P3HT, 51 PTB7-Th and 37 J71 polymer chains were placed into simulation cells of 23.7 × 23.7 × 23.7 nm³ to form the D:A blended systems; the weight ratio of the donor and acceptor is equal to 1:1.2 which corresponds to the experimental value. To simulate the experimental annealing process, MD simulations were performed from 403 K (30 ns) to 300 K (30 ns), where 403 K is the annealing temperature, with a cooling rate from 403 K to 300 K of

10 K/ns. The box size and density data were extracted from the last 10 ns of three molecular dynamics simulations. The final box size of each cell is 10.543 × 10.543 × 10.543, 10.406 × 10.406 × 10.406, and 10.473 × 10.473 × 10.473 nm³ for the P3HT:Y6, PTB7-Th: Y6, and J71:Y6 thin film, respectively. The final system densities were also obtained from the simulations, with values of 1.136, 1.178, and 1.150 g/cm³ for P3HT:Y6, PTB7-Th: Y6, and J71:Y6, respectively. The LINCS algorithm was applied to constrain the covalent bonds with H-atoms[67]. The time step of the simulations was 1.0 fs. All the simulations were performed in periodic boundary conditions with the NPT ensemble. The pressure was coupled at 1 bar by the Parrinello-Rahman method[68], and the temperature was coupled by the Nose-Hoover algorithm[69]. The cut-off of the non-bonded interactions was set to 12 Å. The particle mesh Ewald (PME) method was used to calculate the long-range electrostatic interactions[70]. The graphics and visualization analyses were processed by the Visual Molecular Dynamics (VMD) program[71].

## Capacitance spectroscopy

All devices were encapsulated and tested in air. The complex impedance of the device as a function of frequency was obtained by applying a 10 meV AC signal with frequencies scanned from 4 MHz to 10 kHz from which the total capacitance ($C_{cor}$) of the device was extracted by using Eq. 7 to take into account the presence of series resistance ($R_s$) and parasitic inductance ($L_l$).

$$C_{cor} = -\frac{1}{\omega}\left[\frac{Z'' - \omega L_l}{(Z' - R_s)^2 + (Z'' - \omega L_l)^2}\right] \tag{7}$$

where $Z'$ and $Z''$ are the real and imaginary parts of the complex impedance, respectively, and $\omega$ is the angular frequency. After subtracting the $C_{cor}$ by the geometric capacitance (measured under dark and reverse bias of -1 V), to obtain the total capacitance associated with the active layer only ($C_T$), the chemical capacitance ($C_\mu$) was derived from the saturated $C_T$ at low frequency (Supplementary Fig. 15a). Measurements were carried out under relevant conditions for OPV operation, with devices under 1 sun illumination and constant background bias ($V_{DC}$) scanning from −1 to 0.9 V. Charge carrier density within the active layer ($n$) was obtained by integrating the $C_\mu$ with respect to the corrected applied bias ($V_{cor} = V_{app} - JR_s$) using Eqs. 8 and 9

$$n(V_{cor}) = n_{sat} + \frac{1}{qAL}\int_{V_{sat}}^{V_{cor}} C_\mu dV_{cor} \tag{8}$$

$$n_{sat} = \frac{1}{qAL}C_{sat}(V_0 - V_{sat}) \tag{9}$$

where $C_{sat}$ is the saturated chemical capacitance measured at $V_{sat}$ (−1 V) under 1 sun. $V_0$ is the forward bias at which $J_{ph}$ equals to 0. $L$ is the thickness of the active layer (around 100 nm). Effective charge carrier mobilities at various $V_{cor}$ are obtained via Eq. 10.

$$\mu_{eff}(n, V_{cor}) = \frac{J(V_{cor}) \cdot L}{2qn(V_{cor}) \cdot [V_{cor} - V_{oc}]} \tag{10}$$

## SCLC Measurements

Hole-only or electron-only devices were fabricated as follows: ITO/ PEDOT:PSS/active layer/Au for holes and ITO/ZnO/active layer/Al for electrons. The mobility was extracted by fitting the dark $J$-$V$ curves in the space charge limited current (SCLC) regime using Eq. 11.

$$J = (9/8)\mu\varepsilon r\varepsilon_0 V^2 exp(0.89(V/E_0 L)^{0.5})/L^3 \tag{11}$$

Here, $J$ refers to the dark current density, $\mu$ is hole or electron mobility, $\varepsilon_r$ is the relative dielectric constant of the transport medium, which is equal to 3, $\varepsilon_0$ is the permittivity of free space ($8.85 \times 10^{-12}$ F m$^{-1}$), $V = V_{appl} - V_{bi}$, where $V_{appl}$ is the voltage applied to the device, and $V_{bi}$ is the built-in voltage due to the difference in work function of the two electrodes (for hole-only diodes, $V_{bi}$ is 0.2 V; for electron-only diodes, $V_{bi}$ is 0 V). $E_0$ is the characteristic field describing the extent in which carrier mobility is dependent on electric field[72], $L$ is the thickness of the active layer and was measured by KLA-Tencor Alpha-Step D-600 Stylus Profiler.

## Reporting summary

Further information on research design is available in the Nature Portfolio Reporting Summary linked to this article.

## Data availability

The source data supporting the findings of this study are available from the corresponding authors upon request.

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

## Acknowledgements

The authors acknowledge the financial support from the National Natural Science Foundation of China (No. 52122004) and the Guangdong-Hong Kong-Macao Joint Laboratory for Neutron Scattering Science and Technology (TC2116291). Y. K. is grateful for the funding of the Youth Innovation Promotion Association, CAS (No.2020010). X. C. acknowledges the New Faculty Start-up Grant of the City University of Hong Kong (7200709, 9610547). X. Z. acknowledges the support from the National Natural Science Foundation of China (U21A20101). The work at Arizona was supported by the Office of Naval Research (Award No. N00014-20–1-2110 and N00014-24–1-2114). We thank the beam time and technical supports provided by small-angle neutron scattering sector at China Spallation Neutron Source and BL19U2 beamline at Shanghai Synchrotron Radiation Facility.

## Author contributions

X.L. conceived the idea and supervised the work. G.C., Y.L. and Y.F. contributed equally to this work. G.C. and X.Z. designed the deuteration synthesis. G.C. synthesized *d*-Y6 and *d*-IDIC. Y.L., H.Y. and Y.K. performed GISANS experiments and analyzed the data. L.M., X.C. and J.B. conducted the DFT, MD and PME calculations. G.C., T.L. fabricated and characterized the devices. Y.F. and H.L. conducted capacitance spectroscopy experiments and corresponding analysis. Z.N. and M.-C.T. synthesized *d*-Y7. X.W. contributed to data analysis and revised manuscript preparation. Y. L., G.C., Y.F. and X.L. prepared the manuscript. All the authors provided revisions.

## Competing interests

The authors declare no competing interests.
