## [Peer Review File · Nature Communications]

Deuteration-enhanced neutron contrasts to probe amorphous domain sizes in organic photovoltaic bulk heterojunction filmsREVIEWER COMMENTS

Reviewer #1 (Remarks to the Author):

Cai et. al. used isotope labeling method to measure the amorphous domain size of deuterated Y6 molecules in the blend film of PM6:d-Y6 and experimentally uncovered a structure-property relationship between the short-range aggregation of Y6 molecules and the performance of OPV devices. The authors combined GISAXS and GISANS techniques to evaluate the scattering length densities of polymeric donors and acceptors, which give a large contrast to distinguish the aggregation of acceptors. Although GISANS has been widely utilized in OPV studies (refs. 28-33), the aggregation nature of Y6 in amorphous domains is significant to understanding the role of Y6, as an emerging acceptor, in the development of high-performance solar cells. Thus, I recommend this work to publish in nature communication after the response to the following comments:

- 1) All the data in this manuscript are obtained by GISAXS and GISANS. Although these two techniques are powerful to support the conclusion, would the authors additionally provide AFM phase images to observe and evaluate the phase separation of PM6:d-Y6? To do so, the crystalline and amorphous domain sizes could be measured directly.
- 2) The authors fitted the GISAXS and GISANS intensity profile using equation 1, and provided the information of nanomorphology in PM6:d-Y6 film. In line 226-227, they claim that GISAXS profile gave “the correlation length (ξ) of the amorphous intermixing phase and the size of the crystalline acceptor domain.” In my opinions, both the PM6 and d-Y6 are crystalline in the blend films, so why only crystalline domain of Y6 acceptor can be evaluated by fitting the equation?
- 3) Would the authors explain the reason to choose a PM6:Y6 BHJ blend as the active layers in OPV devices instead of PM6:d-Y6? Since all the information on the nanomorphology of blend films comes from PM6:d-Y6, the corresponding devices are better to be tested for data consistency, although the performance of two devices comprising PM6:Y6 and PM6:d-Y6 may be identical (as shown in Fig. S8b)

Others:

- 4) line 57, the “research” should be plural so “has” is used wrongly in grammar.
- 5) line 91, the “challenghg” is a typo.
- 6) line 103, “not” in the term of “it originates not from crystalline phases...” is inappropriately expressed in grammar.
- 7) line 129, the “aggregation” should be singular.
- 8) line 137, “Supplementary Fig. 6” should be “Supplementary Fig. 7”.
- 9) line 202, the correspondence of fig. 3a and 3b to GISANS and GISAXS is an error.

Reviewer #2 (Remarks to the Author):

The main novelty of this work is that the authors deuterated the non-fullerene acceptor (NFA) Y6 to overcome the lack of neutron contrast in OPV films based on NFAs. This allowed them to study the nanomorphology of the OPV films using GISANS. This work is merely incremental, and it lacks the originality needed to be published in this journal.

In the abstract the authors say that their work uncovers "for the first time" the amorphous nanomorphology of OPV films. This is not correct – this has been addressed before either using neutron scattering or Resonant Soft X-ray Scattering (RSoXS) techniques.

Neutron scattering (SANS and Neutron reflectivity) have been previously used extensively to study fullerene-based OPVs and the authors are advised to improve their literature survey and include some important references that are missing. For example, these below among many others.

Wienhold, K.S., "Organic solar cells probed with advanced neutron scattering techniques" Applied Physics Letters, 116, 120504 (2020)

Zhang, Y. et al, "Understanding and controlling morphology evolution via DIO plasticization in PffBT4T-2OD/PC71BM devices", Scientific Reports 7, 44269 (2017)

Chen D. "P3HT/PCBM Bulk Heterojunction Organic Photovoltaics: Correlating Efficiency and Morphology", Nano Lett. 2011, 11, 2, 561–567

Bernardo, G. et al "Impact of 1,8-diiodooctane on the morphology of organic photovoltaic (OPV) devices – A Small Angle Neutron Scattering (SANS) study", Polymer Testing 82 (2020) 106305

Reviewer #3 (Remarks to the Author):

The authors used a combination of neutron scattering - in terms of GISANS technique - and deuteration to probe the nanomorphology of the PM6:d-Y6 blend film. The deuteration was selectively done for the NFA Y6 (d-Y6). The measurements were underpinned by MD calculations. The study highlighted a specific short-range aggregation of Y6 - as an intrinsic feature of this NFA, that was concluded to be a key contributor to the high photovoltaic performance of most Y6-containing organic bulk heterojunction systems. Indeed, the authors studied other polymers than PM6, and have supported the neutron data by measurements using different x-ray and light-based techniques.

The manuscript is dense, organized and well written. The topic, techniques and findings warrant publication in Nature Communications after considering the following comment.

The only concern readers, who might be interested in applying neutron scattering and associated deuteration in their research on OPVs, would be to have some more targeted references on the application of neutron scattering and deuteration to study morphology and local properties, including

dynamics, beyond GISANS by including other neutron-based techniques. In this context, the authors are invited to enrich the introduction and the reference list by considering recent works on binary and ternary blend, either with acceptors being fullerenes (PCBM, etc) or NFAs (O-IDTBR, etc).

Reviewer #4 (Remarks to the Author):

Deuteration-enhanced neutron is cleverly applied to GISANS test, and the unique short-range aggregation behavior of Y6 in the D/A blending region is effectively detected, which also confirms the experimental results of other research groups. Prof Lu and co-workers provide a novel method for detecting the aggregate state of non-crystalline domain in active layer of OSCs. However, SANS is already a conventional and mature characterization technique, and deuterated materials have also been applied to SANS. In addition, the PM6:Y6 device in the paper is a very conventional, although the authors claim that directionally deuterated SANS technology offers opportunities for complex active layer systems and intermolecular interaction mechanisms. The innovation of this article is insufficient. The authors did not validate the proposed large short-range aggregation of acceptors in the D/A blend region as a key to recent high-performance organic solar cells and provide insights for improving the efficiency or stability of organic solar cells, so it is not recommended for publication in Nature Communication.

1. Prof. Ye has demonstrated that deuterated solvents can improve the performance of organic solar cells (Aggregate 2023, 4, e289). d-Y6 shows similar photoelectric properties to Y6 and does not affect the morphology of the active layer. When blended with PM6 to make a device, how does the performance of d-Y6 compare to that of Y6?
2. To understand the formation mechanism of Y6 or d-Y6 aggregates, the single crystal packing is more solid result compared to molecular dynamics simulation. The single crystal of d-Y6 need to be provided.
3. The use of deuterated materials is costly and does not improve efficiency, has little application and is too costly, so the significance of this study is yet to be determined.
4. ξ in line 310 is in difference from in Table 1. Please check it.
5. The scattering lengths of hydrogen and deuterium have opposite symbols, which allows SANS to use this contrast advantage to selectively label different parts of the study target system. Whether hydrogen in PM6 and deuterium in Y6 can be detected to more deeply interpret the molecular state in the amorphous domain ?
6. The author has proved that short-range aggregation behavior is unique to Y6. Whether other high-performance Y-series acceptors have similar behavior? The authors need to extend the corresponding experiment for d-L8-BO and d-BO-4Cl to support their conclusion.
7. The ^{13}C NMR and elemental analysis of new compounds need to be provided.

**Responses to comments from reviewers**

We would like to thank the reviewers and the editor for their valuable time and
comments, which have greatly helped to improve the quality of our paper. We have
addressed the comments from each reviewer point by point and revised the manuscript
accordingly. Our detailed responses are shown in blue and the corresponding revisions
in both the manuscript and supplementary information are indicated with yellow
highlights.

**Reviewer #1:**

Comments:

Reviewer #1 (Remarks to the Author):

Cai et. al. used isotope labeling method to measure the amorphous domain size of
deuterated Y6 molecules in the blend film of PM6:d-Y6 and experimentally uncovered
a structure-property relationship between the short-range aggregation of Y6 molecules
and the performance of OPV devices. The authors combined GISAXS and GISANS
techniques to evaluate the scattering length densities of polymeric donors and
acceptors, which give a large contrast to distinguish the aggregation of acceptors.
Although GISANS has been widely utilized in OPV studies (refs. 28-33), the
aggregation nature of Y6 in amorphous domains is significant to understanding the role
of Y6, as an emerging acceptor, in the development of high-performance solar cells.
Thus, I recommend this work to publish in nature communication after the response to
the following comments:

**Reply:** We thank the reviewer for the overall positive comments.

1) All the data in this manuscript are obtained by GISAXS and GISANS. Although
these two techniques are powerful to support the conclusion, would the authors
additionally provide AFM phase images to observe and evaluate the phase separation
of PM6: d-Y6? To do so, the crystalline and amorphous domain sizes could be measured

directly.

**Reply:** We thank the reviewer for this constructive comment. To complement our
scattering measurements, we measured tapping-mode AFM topography images for
PM6:Y6 and PM6:*d*-Y6 blend films processed under three different conditions (as-cast,
CF-opt, and CB-opt, as mentioned in the main text). AFM images of PM6:Y6 blend
films are similar to those of PM6:*d*-Y6 under each processing condition. The root-
mean-square (RMS) roughness of blend films increases from as-cast to CF-opt, and
then to CB-opt, indicating the enhanced molecular aggregation/crystallization induced
by prolonged film drying times. Since topography images partially reflect phase-
separated structures in blend films, we observed spherical agglomeration in CB-opt
films with sizes of tens of nm, consistent with the GISAXS fitting results presented in
the main text. However, we note that the short-range Y6 aggregates probed using
GISANS cannot be identified from AFM images as their sizes (5-10 nm) are below the
resolution of the tip we used (Tap300Al-G, Budget Sensor), which is around 10 nm.
We added the AFM topography images to **Supplementary Fig. 20**.

**Supplementary Fig. 20** AFM topography images of PM6:Y6 (a-c) and PM6: *d*-Y6 (d-

f) blend films processed under different conditions mentioned in the main text. The
root-mean-square (RMS) roughness of heights is labelled at the inset for each image.

We added the following sentences to line 293, page 16 of the main text.

Tapping-mode atomic-force microscopy (AFM) was also applied to measure the
surface topography of PM6:Y6 and PM6:*d*-Y6 films processed under the three
aforementioned conditions as shown in **Supplementary Fig. 20**. The spherical
agglomerates with sizes of around 50 nm in CB-opt is also visible from AFM, further
supporting our GISAXS fitting results. However, AFM cannot identify the short-range
Y6 aggregates (5-10 nm) observed from GISANS measurements as their sizes are
below the resolution limit of the AFM tip we used (around 10 nm, see **Methods**).

We added the following sentences to line 470, page 26 of the main text.

**AFM measurements** JPK NanoWizard NanoOptics atomic force microscope (AFM)
with a Tap300A1-G tip (40 N/m) was used for topography characterization of blend
films. Measurements were carried out using non-contact mode with a piezoelectric tip
oscillating at a fixed frequency (300 kHz) above the sample surface. For each
measurement, tip was scanned over 256 pixels across a 2 μm range at a rate of 1 Hz. A
$2 \times 2 \mu\text{m}$ image was obtained for each sample. The root-mean-square (RMS) height
fluctuations were obtained using the JPKSPM Data Processing software package.

2) The authors fitted the GISAXS and GISANS intensity profile using equation 1, and
provided the Information of nanomorphology in PM6: *d*-Y6 film. In line 226-227, they
claim that GISAXS profile gave “the correlation length (ξ) of the amorphous
intermixing phase and the size of the crystalline acceptor domain.” In my opinions, both
the PM6 and *d*-Y6 are crystalline in the blend films, so why only crystalline domain of
Y6 acceptor can be evaluated by fitting the equation?

**Reply:** We thank the reviewer for pointing out the potential confusion. The scattering
contrast of a blend film under X-ray arises from the different SLDs of constituent
materials. For PM6 and Y6 with similar carbon/hydrogen compositions, this is mainly
induced by their different degree of crystallinity in blend films. Based on our previous
experience, NFA small molecules typically show stronger scattering of X-ray than D:A
polymer donor due to their stronger degree of crystallinity [*Nat. Commun.* 12, 6226
(2021)]. Furthermore, in the blend film, it has been shown that Y6 tend to form denser
and stronger crystals in blend films, substantially reducing the crystallinity of PM6 [*Adv*
*Mater.*, e2302005 (2023)]. Therefore, for the fitting of our GISAXS results, we choose
the DAB model to account for the amorphous region of the PM6/*d*-Y6 mixtures and the
fractal model to account for the *d*-Y6 crystalline domains.

3) Would the authors explain the reason to choose a PM6:Y6 BHJ blend as the active
layers in OPV devices instead of PM6: *d*-Y6? Since all the information on the
nanomorphology of blend films comes from PM6: *d*-Y6, the corresponding devices are
better to be tested for data consistency, although the performance of two devices
comprising PM6:Y6 and PM6: *d*-Y6 may be identical (as shown in Fig. S8b)

Others:

**Reply:** We thank the reviewer for the insightful suggestions. The reason we chose to
display the data of PM6:Y6 in the main text is to emphasis that the GISANS results we
obtained from PM6:*d*-Y6 are also applicable to normal non-deuterated devices.
Deuteration is expensive, we have no intention to apply deuteration to improve the

device performance, which is also not feasible for mass production. We just want to
employ it as a powerful labeling tool for the research of OPV morphology.

To understand the impact of deuteration on morphology, optoelectronic properties, and
device performance, we have shown in **Fig. 1c, d**, and **Supplementary Fig. 10-12** that
deuteration does not change the morphological of Y6 molecules in neat and blend films
and thus does not notably impact on and optoelectronic properties and device
performance. We further fabricated a new batch of PM6:Y6 and PM6:*d*-Y6 devices
under three different processing conditions mentioned in the main text to compare their
performances as shown in **Supplementary Fig. 17** and summarized their detailed
device parameters in **Supplementary Table 3**.

**Supplementary Fig. 17** Typical J-V curves of the devices based on PM6:Y6 and
PM6:*d*-Y6 under three different conditions (**a** CF-ac, **b** CF-opt, **c** CB-opt) mentioned
in the main text.

**Supplementary Table 3** A summary of device characteristics of PM6:Y6 and PM6:*d*-
Y6 systems under three different processing conditions mentioned in the main text.

Blends	V_{oc} (V) ^d	J_{sc} (mA·cm ⁻²) ^d	FF (%) ^d	PCE (%) ^d
PM6: Y6 ^a	0.865 (0.861±0.003)	26.9 (26.7±0.2)	69.7 (69.0±0.5)	16.3 (15.9±0.3)
PM6: d-Y6 ^a	0.856 (0.855±0.003)	27.1 (26.7±0.3)	67.2 (66.8±0.5)	15.6 (15.2±0.2)
PM6: Y6 ^b	0.853 (0.850±0.002)	28.0 (27.7±0.5)	74.4 (74.1±0.4)	17.8 (17.4±0.3)
PM6: d-Y6 ^b	0.846 (0.843±0.002)	27.7 (27.7±0.2)	75.1 (74.7±0.3)	17.6 (17.4±0.2)
PM6: Y6 ^c	0.806 (0.798±0.006)	25.5 (24.7±0.6)	72.2 (71.5±0.5)	14.8 (14.1±0.5)
PM6: d-Y6 ^c	0.807 (0.807±0.003)	24.9 (24.6±0.5)	72.1 (72.0±0.4)	14.5 (14.3±0.2)

117 ^a As-cast w/o CN & TA. ^b CF-opt (w/ 0.5% CN & TA 90°C 5min). ^c CB-opt (w/ 0.5%

CN & TA 90°C 5min). ^d Average values with their standard deviations (in parentheses)

are obtained from 5 independent devices.

PM6:Y6 and PM6:d-Y6 show similar performances in terms of all three photovoltaic

parameters under each processing condition, confirming the validity of our analysis.

We added the following sentences on line 272, page 15 in the main text.

Under each processing condition, PM6:Y6 and PM6:d-Y6 devices show similar

performance as shown in **Supplementary Fig. 17** and **Supplementary Table 3**,

suggesting that the GISANS results obtained from PM6:d-Y6 blend films are relevant

for normal non-deuterated devices as well.

We updated the device performances in Table 1 of the main text.

**Table 1** A summary of fitted domain sizes and device characteristics of PM6:Y6

systems.

Samples ^a	ξ (nm)	$2R_{gc}$ (nm)	$2R_{ga}$ (nm)	V_{oc} ^b (V)	J_{sc} ^b (mA cm ⁻²)	FF ^b (%)	PCE ^b (%)
CF-ac	45.6	33.1	7.8	0.865 (0.861±0.003)	26.9 (26.7±0.2)	69.7 (69.0±0.5)	16.3 (15.9±0.3)

CF-opt	49.4	29.4	10.7	0.853 (0.850±0.002)	28.0 (27.7±0.5)	74.4 (74.1±0.4)	17.8 (17.4±0.3)
CB-opt	N/A	58.1 ^c	7.5	0.806 (0.798±0.006)	25.5 (24.7±0.6)	72.2 (71.5±0.5)	14.8 (14.1±0.5)

134 ^a D/A = 1/1.2 (w/w). ^b Average values with their standard deviations (in parentheses)
 are obtained from 15 independent devices. ^c The 2R_{gc} of the CB-opt film is obtained
 from the GISAXS fitting.

4) line 57, the “research” should be plural so “has” is used wrongly in grammar.

**Reply:** Thanks for your correction; we have already made the necessary changes.

5) line 91, the “challengg” is a typo.

**Reply:** Thanks for your correction; we have already made the necessary changes.

6) line 103, “not” in the term of “it originates not from crystalline phases...” is
 inappropriately expressed in grammar.

**Reply:** Thanks for your correction; we have already made the necessary changes.

7) line 129, the “aggregation” should be singular.

**Reply:** Thanks for your correction; we have already made the necessary changes.

8) line 137, “Supplementary Fig. 6” should be “Supplementary Fig. 7”.

**Reply:** Thanks for your correction; we have already made the necessary changes.

9) line 202, the correspondence of fig. 3a and 3b to GISANS and GISAXS is an error.

**Reply:** Thanks for your correction; we have already made the necessary changes.

**Reviewer #2:**

Comments:

The main novelty of this work is that the authors deuterated the non-fullerene acceptor
(NFA) Y6 to overcome the lack of neutron contrast in OPV films based on NFAs. This
allowed them to study the nano-morphology of the OPV films using GISANS. This
work is merely incremental, and it lacks the originality needed to be published in this
journal.

In the abstract the authors say that their work uncovers "for the first time" the
amorphous nanomorphology of OPV films. This is not correct – this has been addressed
before either using neutron scattering or Resonant Soft X-ray Scattering (RSoXS)
techniques.

Neutron scattering (SANS and Neutron reflectivity) have been previously used
extensively to study fullerene-based OPVs and the authors are advised to improve their
literature survey and include some important references that are missing. For example,
these below among many others.

Wienhold, K.S., "Organic solar cells probed with advanced neutron scattering
techniques" Applied Physics Letters, 116, 120504 (2020)

Zhang, Y. et al, "Understanding and controlling morphology evolution via DIO
plasticization in PffBT4T-2OD/PC71BM devices", Scientific Reports 7, 44269 (2017)

Chen D. "P3HT/PCBM Bulk Heterojunction Organic Photovoltaics: Correlating
Efficiency and Morphology", Nano Lett. 2011, 11, 2, 561–567

Bernardo, G. et al "Impact of 1,8-diiodooctane on the morphology of organic
photovoltaic (OPV) devices – A Small Angle Neutron Scattering (SANS) study",
Polymer Testing 82 (2020) 106305

**Reply:** We appreciate the reviewer's insightful comments and suggestions. We believe
there may be a misunderstanding regarding our definition of "amorphous

nanomorphology of organic photovoltaic thin films”. In our work, it refers to
morphology within the **amorphous donor:acceptor intermixed domains**. Through a
thorough literature search, we have ensured that this morphology has not been detected
previously.

Next, we agree that neutron scattering techniques, including SANS and Neutron
reflectivity, have been previously used extensively to study **fullerene-based OPVs**. In
the introduction section of the original manuscript (line 76-90, page 5), we wrote:

Small-angle neutron scattering (SANS) and neutron reflectivity (NR) have been
employed to investigate the nanostructure of organic solar cells based on fullerene
acceptors since the dominant carbon components in fullerene derivatives provide
sufficient SLD contrast relative to organic donor materials^{28,29}. In a study by Dadmun
*et al.*, SANS was utilized for the first time to reveal the miscibility of PCBM in P3HT,
the average PCBM domain size, and the interfacial area between PCBM and the P3HT-
rich phase³⁰. The results of SANS experiments conducted by Nedoma *et al.* indicated
that the phase separation between PCBM and P3HT could be controlled by device
fabrication conditions³¹. In 1999, P. Müller-Buschbaum *et al.* developed grazing
incidence small-angle neutron scattering (GISANS) to enhance the signal-to-noise ratio
and scattering volume with the grazing incidence geometry³². In subsequent studies,
Matthias *et al.* first applied GISANS in OPV studies and investigated the phase
separation and molecular intermixing in the P3HT/PC₆₁BM bulk heterojunction³³. In
2018, W. Wang *et al.* used time-of-flight (TOF)-GISANS in the P3HT:PC₆₁BM bulk
heterojunction thin film and quantitatively determined the molecular miscibility
between P3HT and PC₆₁BM as well as the depth-dependent morphology changes
induced by additives²⁸.

We also thank the reviewer for reminding us of more relevant literature. Particularly,

the paper [*Chem. Phys.* 427, 142-146 (2013).] introduced a new quasi-elastic neutron
scattering (QENS) to monitor the motions of the polymer side-chains of P3HT which
were slowed down upon addition and further crystallization of the PCBM molecules.
We have added all references mentioned by the reviewer to the following paragraph on
page 5 of the revised main text.

Small-angle neutron scattering (SANS), neutron reflectivity (NR) and quasi-elastic
neutron scattering (QENS) have been employed to investigate the nanostructure,
dynamic fluctuations of OPV active layers composed of fullerene acceptors since the
dominant carbon components in fullerene derivatives provide sufficient SLD contrast
relative to organic donor materials^{28,29}. In the study by Dadmun *et al.*, SANS was
utilized for the first time to reveal the miscibility of PCBM in P3HT, the average PCBM
domain size, and the interfacial area between PCBM and the P3HT-rich phases³⁰. The
results of SANS experiments conducted by Nedoma *et al.* indicated that the phase
separation between PCBM and P3HT could be controlled by device fabrication
conditions³¹. More following research also demonstrated the nanomorphology of
fullerene based organic solar cells by SANS³²⁻³⁴. In terms of dynamic information,
quasi-elastic neutron scattering (QENS) was utilized to monitor the motions of the side-
chains of P3HT polymers which were slowed down upon addition and further
crystallization of the PCBM molecules^{35,36}. In 1999, P. Müller-Buschbaum *et al.*
developed grazing incidence small-angle neutron scattering (GISANS) to enhance the
signal-to-noise ratio and scattering volume with the grazing incidence geometry³⁷.
Matthias *et al.* first applied GISANS in OPV studies and investigated the phase
separation and molecular intermixing in the P3HT/PC₆₁BM bulk heterojunction³⁸.
Subsequently, this technique was applied by Guo *et al.* to study the impact of alcohol
post treatment on inner phase structure of PTB7:PC₇₁BM blend films³⁹.

In summary, all these references are focused on using transmission-mode SANS or

GISANS to probe the morphology of fullerene-based blend films, taking advantage of
the much stronger scattering contrast between fullerene molecules and polymer donors
under neutron beam. The same methodology cannot be directly transferred to
polymer:non-fullerene acceptor (NFA) blend films where donor and acceptor molecules
have similar carbon: hydrogen ratios and thus similar SLDs under both X-ray and
neutron. This challenge motivates us to combine targeted deuteration and GISANS to
further probe morphology in these NFA-based blend systems. Therefore, the references
mentioned by the reviewer do not undermine the originality of our work but rather
highlighting the innovation in our approach.

We added the following works to our reference list.

Bernardo, G. *et al.* Impact of 1,8-diiodooctane on the morphology of organic
photovoltaic (opv) devices – a small angle neutron scattering (sans) study.
*Polymer Testing* **82** (2020).

Chen, D., Nakahara, A., Wei, D., Nordlund, D. & Russell, T. P. P3ht/pcbm bulk
heterojunction organic photovoltaics: Correlating efficiency and morphology.
*Nano Lett* **11**, 561-567 (2011).

Zhang, Y. *et al.* Understanding and controlling morphology evolution via dio
plasticization in pffbt4t-2od/pc(71)bm devices. *Sci Rep* **7**, 44269 (2017).

Paternó, G., Cacialli, F. & García-Sakai, V. Structural and dynamical
characterization of p3ht/pcbm blends. *Chem. Phys.* **427**, 142-146 (2013).

Wienhold, K. S., Jiang, X. & Müller-Buschbaum, P. Organic solar cells probed
with advanced neutron scattering techniques. *Appl. Phys. Lett.* **116** (2020).

Guo, S., Cao, B., Wang, W., Moulin, J.-F. & Müller-Buschbaum, P. Effect of
alcohol treatment on the performance of ptb7:Pc71bm bulk heterojunction solar
cells. *ACS Appl. Mater. Interfaces* **7**, 4641-4649 (2015).

**Reviewer #3:**

Comments:

Reviewer #3 (Remarks to the Author):

The authors used a combination of neutron scattering - in terms of GISANS technique

- and deuteration to probe the nanomorphology of the PM6:d-Y6 blend film. The
deuteration was selectively done for the NFA Y6 (d-Y6). The measurements were
underpinned by MD calculations. The study highlighted a specific short-range
aggregation of Y6 - as an intrinsic feature of this NFA, that was concluded to be a key
contributor to the high photovoltaic performance of most Y6-containing organic bulk
heterojunction systems. Indeed, the authors studied other polymers than PM6, and have
supported the neutron data by measurements using different x-ray and light-based
techniques.

The manuscript is dense, organized and well written. The topic, techniques and findings
warrant publication in Nature Communications after considering the following
comment.

The only concern readers, who might be interested in applying neutron scattering and
associated deuteration in their research on OPVs, would be to have some more targeted
references on the application of neutron scattering and deuteration to study morphology
and local properties, including dynamics, beyond GISANS by including other neutron-
based techniques. In this context, the authors are invited to enrich the introduction and
the reference list by considering recent works on binary and ternary blend, either with
acceptors being fullerenes (PCBM, etc) or NFAs (O-IDTBR, etc).

**Reply:** We really appreciate the reviewer's positive comments and insightful
suggestions. Following the reviewer's suggestion, we have enriched introduction and
the reference list by including more important works on ternary OPVs incorporating
either fullerene-based acceptors [*J. Mater. Chem. A* 7, 20713-20722 (2019)] or NFAs
like IDTBR [*Nat. Mater.* 16, 363-369 (2017).] and Y6 [*Dyes Pigm.* 181, 108613 (2020)]
as the third component. We pointed out that despite the ternary (and quaternary) has
improved the efficiency and stability of OPVs, the exact microscopic origin remains
under debate [*Nat. Rev. Mater.* 8, 456-471 (2023). & *Nat. Energy* 8, 978-988 (2023)].
This can be largely attributed to the similar chemical structures between donor and

acceptor materials (in case of NFAs) or the fact that the amount of dopant added is small
compared to the host materials (in case of fullerenes) that renders morphology
characterization challenging. This dilemma could potentially be resolved by combining
deuteration labelling and GISANS, which will be the focus of our future work.

We added the following sentences to line 105 of page 6.

The same issue also hinders the full characterization of ternary (and quaternary) blend
films because the dopants used typically have similar chemical structures with host
materials or the amount of dopants added is small. Therefore, although ternary
strategies incorporating both fullerene⁴⁰ and NFAs (e.g. IDTBR⁴¹ and Y6⁴²) have
proved effective in improving the efficiency and stability of OPVs, the exact
microscopic origin remains controversial^{43,44}.

In addition to SANS and GISANS, we also added a brief introduction on quasi-elastic
neutron scattering (QENS) on line 91 of page 5.

In terms of dynamic information, quasi-elastic neutron scattering (QENS) was utilized
to monitor the motions of the polymer side-chains of P3HT which were slowed down
upon addition and further crystallization of the PCBM molecules^{35,36}.

We have also reviewed a few more examples of previous studies combining deuteration
and neutron scattering in other fields beyond organic photovoltaics on line 111 of page
6.

This technique has been previously applied to study e.g., the structure of conducting
polymers⁴⁵ and biological macromolecules⁴⁶, yet it has not been applied to probe the

morphology of OPV active layers.

The following references were added to the main text.

35 Paternó, G., Cacialli, F. & García-Sakai, V. Structural and dynamical
characterization of p3ht/pcbm blends. *Chem. Phys.* **427**, 142-146 (2013).

36 Wienhold, K. S., Jiang, X. & Müller-Buschbaum, P. Organic solar cells probed
with advanced neutron scattering techniques. *Appl. Phys. Lett.* **116** (2020).

Pan, M.-A. *et al.* 16.7%-efficiency ternary blended organic photovoltaic cells
with pcbm as the acceptor additive to increase the open-circuit voltage and
phase purity. *J. Mater. Chem. A* **7**, 20713-20722 (2019).

Baran, D. *et al.* Reducing the efficiency–stability–cost gap of organic
photovoltaics with highly efficient and stable small molecule acceptor ternary
solar cells. *Nat. Mater.* **16**, 363-369 (2017).

Jiang, B.-H. *et al.* The role of y6 as the third component in fullerene-free ternary
organic photovoltaics. *Dyes Pigm.* **181**, 108613 (2020).

Günther, M. *et al.* Models and mechanisms of ternary organic solar cells. *Nature*
*Reviews Materials* **8**, 456-471 (2023).

Wang, Y. *et al.* Origins of the open-circuit voltage in ternary organic solar cells
and design rules for minimized voltage losses. *Nature Energy* **8**, 978-988 (2023).

Shao, M. *et al.* The isotopic effects of deuteration on optoelectronic properties
of conducting polymers. *Nat Commun* **5**, 3180 (2014).

Jeffries, C. M. *et al.* Preparing monodisperse macromolecular samples for
successful biological small-angle x-ray and neutron-scattering experiments. *Nat*
*Protoc* **11**, 2122-2153 (2016).

**Reviewer #4:**

Comments:

Reviewer #4 (Remarks to the Author):

Deuteration-enhanced neutron is cleverly applied to GISANS test, and the unique short-
range aggregation behavior of Y6 in the D/A blending region is effectively detected,
which also confirms the experimental results of other research groups. Prof Lu and co-
workers provide a novel method for detecting the aggregate state of non-crystalline
domain in active layer of OSCs. However, SANS is already a conventional and mature
characterization technique, and deuterated materials have also been applied to SANS.

In addition, the PM6:Y6 device in the paper is a very conventional, although the authors
claim that directionally deuterated SANS technology offers opportunities for complex
active layer systems and intermolecular interaction mechanisms. The innovation of this
article is insufficient. The authors did not validate the proposed large short-range
aggregation of acceptors in the D/A blend region as a key to recent high-performance
organic solar cells and provide insights for improving the efficiency or stability of
organic solar cells, so it is not recommended for publication in Nature Communication.

**Reply:** We appreciate the thoughtful comments provided by the reviewer and have
carefully addressed them point-by-point as detailed in the following response. In
response to the observation that SANS and deuteration have been previously applied
together, we wish to highlight the unique contribution of our work. While this
combination has been utilized in other fields, for instance, biology systems, our work
represents the first application to investigate the amorphous nanomorphology in organic
photovoltaic thin films. In addition, the reason we chose to use the conventional system
of PM6:Y6 to demonstrate this methodology is because we want to highlight its
generality, while recent advancements on OPV performance often utilize Y-series
acceptors. Following the reviewer's suggestion, in the revised manuscript, we further
deuterate another Y-series molecule-Y7 to demonstrate the generality of our results.

2. Prof. Ye has demonstrated that deuterated solvents can improve the performance of
organic solar cells (Aggregate 2023, 4, e289). d-Y6 shows similar photoelectric
properties to Y6 and does not affect the morphology of the active layer. When blended
with PM6 to make a device, how does the performance of d-Y6 compare to that of Y6?

**Reply:** We thank the reviewer for reminding us of this important work. In Prof. Ye's
work, the deuterium substitution is on **solvent molecules**, not **acceptors**. The
deuterated solvent reduces their molar volume and polarizability by decreasing the

length and dipole moment of C-H bonds, resulting in suppressed solubility parameter
(δ) of active layer materials in deuterated solvents, and therefore increase the
crystallinity of the casted film. We did not observe similar device performance
improvement with the deuteration of Y6. Prof. Ye and co-workers recently

In **Supplementary Fig. 11** and **12b**, we have shown that PM6:Y6 and PM6:*d*-Y6 blend
films have similar crystal structures and photovoltaic performances. To further stress
this point, we also compared PM6:Y6 and PM6:*d*-Y6 devices fabricated under three
different conditions (as-cast, CF-opt, and CB-opt). Under each condition, the PM6:*d*-
Y6 device shows similar performance in terms of all three photovoltaic parameters (FF ,
J_{sc} , and V_{oc}) with the corresponding PM6:Y6 device. This confirms that deuteration of
Y6 has negligible impact on their morphological and optoelectronic properties in both
neat and blend films. We added the J-V curves to **Supplementary Fig. 17** and detailed
device characteristics to **Supplementary Table 3**.

**Supplementary Fig. 17** Typical J-V curves of the devices based on PM6:Y6 and
PM6:*d*-Y6 under three different conditions (**a** CF-ac, **b** CF-opt, **c** CB-opt) mentioned
in the main text.

**Supplementary Table 3** A summary of device characteristics of PM6:Y6 and PM6:*d*-
Y6 systems under three different processing conditions mentioned in the main text.

Blends	V_{oc} (V) ^d	J_{sc} (mA·cm ⁻²) ^d	FF (%) ^d	PCE (%) ^d
PM6: Y6 ^a	0.865 (0.861 ± 0.003)	26.9 (26.7 ± 0.2)	69.7 (69.0 ± 0.5)	16.3 (15.9 ± 0.3)
PM6: d-Y6 ^a	0.856 (0.855 ± 0.003)	27.1 (26.7 ± 0.3)	67.2 (66.8 ± 0.5)	15.6 (15.2 ± 0.2)
PM6: Y6 ^b	0.853 (0.850 ± 0.002)	28.0 (27.7 ± 0.5)	74.4 (74.1 ± 0.4)	17.8 (17.4 ± 0.3)
PM6: d-Y6 ^b	0.846 (0.843 ± 0.002)	27.7 (27.7 ± 0.2)	75.1 (74.7 ± 0.3)	17.6 (17.4 ± 0.2)
PM6: Y6 ^c	0.806 (0.798 ± 0.006)	25.5 (24.7 ± 0.6)	72.2 (71.5 ± 0.5)	14.8 (14.1 ± 0.5)
PM6: d-Y6 ^c	0.807 (0.807 ± 0.003)	24.9 (24.6 ± 0.5)	72.1 (72.0 ± 0.4)	14.5 (14.3 ± 0.2)

412 ^a As-cast w/o CN & TA. ^b CF-opt (w/ 0.5% CN & TA 90°C 5min). ^c CB-opt (w/ 0.5%

CN & TA 90°C 5min). ^d Average values with their standard deviations (in parentheses)

are obtained from 5 independent devices.

We also added the following sentences to line 272, page 15 of the main text.

Under each processing condition, PM6:Y6 and PM6:d-Y6 devices show similar

performance as shown in **Supplementary Fig. 17** and **Supplementary Table 3**,

suggesting that the GISANS results obtained from PM6:d-Y6 blend films are relevant

for normal non-deuterated devices as well.

2. To understand the formation mechanism of Y6 or d-Y6 aggregates, the single crystal

packing is more solid result compared to molecular dynamics simulation. The single

crystal of d-Y6 need to be provided.

**Reply:** Thanks a lot for the suggestion. Single-crystal analysis is a powerful tool to

capture crystalline packing motifs of conjugated molecules in their thermodynamically

stable states. However, this work is focusing on the **amorphous** morphology but not

the crystalline packing of Y6. Furthermore, it cannot predict the formation of short-

range Y6 aggregates within the **blend** film with polymer donors cast by spin coating.

Therefore, we believe the single crystal analysis of d-Y6 is not necessary here and MD

simulation is a more valid method to complement our experimental results to

understand the formation mechanism of short-range Y6 aggregates in thin films formed
through non-equilibrium kinect process.

3. The use of deuterated materials is costly and does not improve efficiency, has little
application and is too costly, so the significance of this study is yet to be determined.

**Reply:** We thank the reviewer for raising this important concern. Yes, the deuteration
of conjugated small molecules or even polymers is costly and it does not notably affect
the performance of OPV devices as we have shown in the main text. We emphasis that
our goal is to apply deuteration substitution as an effective labelling technique to
complement GISANS measurements so that we could reveal the previously hidden
short-range structure and establish robust processing-structure-performance
relationships for OPVs. Therefore, we only need to synthesis a relatively small amount
of deuterated materials (e.g. several tens of mg) to help us with morphology
characterizations so the cost is affordable.

4. ξ in line 310 is in difference from in Table 1. Please check it.

**Reply:** Thanks for your correction; we have already made the necessary changes.

5. The scattering lengths of hydrogen and deuterium have opposite symbols, which
allows SANS to use this contrast advantage to selectively label different parts of the
study target system. Whether hydrogen in PM6 and deuterium in Y6 can be detected to
more deeply interpret the molecular state in the amorphous domain?

**Reply:** We thank the reviewer for this inspiring suggestion. So far, we are making use
of the enhanced contrast to probe the statistical averaged domain sizes of the crystalline
and amorphous Y6. If we want to obtain the molecular state in the amorphous domain,
we may need to design more delicate deuteration schemes on different functional
groups for PM6 and Y6 and compare the scattering difference of different combinations
to extract the detailed molecular state, which is way beyond the scope of this work.

6. The author has proved that short-range aggregation behavior is unique to Y6.

Whether other high-performance Y-series acceptors have similar behavior? The authors
need to extend the corresponding experiment for d-L8-BO and d-BO-4Cl to support
their conclusion.

**Reply:** We thank the reviewer for the valuable suggestions. To prove the generality of
our findings for other Y-series NFAs, in the revised manuscript, we have synthesized a
new Y-series NFA - *d*-Y7, which shares the same backbone with Y6 but with chlorinated
end groups [*Nat. Commun.* 10, 1-8 (2019).]. The GISANS results confirmed that *d*-Y7
shows a higher SLD of $5.37 \times 10^{-6} \text{ \AA}^{-2}$ compared to Y7 ($1.77 \times 10^{-6} \text{ \AA}^{-2}$). Encouragingly,
the scattering feature associated with short-range aggregates was also observed in the
blend film of PM6:*d*-Y7 with characteristic length of around 5.4 nm. Considering the
similar molecular structures of Y6 and Y7, we anticipate that the prerequisite for
forming short-range aggregates is the ability to maintain high backbone planarity. It has
been shown previously that the beta side chains attached to the outer core group of Y6,
regardless of their chain lengths and shapes, are crucial to suppress dihedral angle
between core and end groups [*Energy Environ. Sci.* 13, 2422-2430 (2020). & *Nat. Rev.*
*Mater.* 8, 839-852 (2023).]. Therefore, our conclusion should be general for most Y-
series NFAs with side chains attached to this position.

We added the results of ^1H and ^{13}C NMR spectroscopy to the Supporting Information.

**Supplementary Fig. 5** ^1H NMR spectrum of *d*-Y7.

**Supplementary Fig. 6** ^{13}C NMR spectrum of *d*-Y7.

***d*-Y7** The material was synthesized following the same route as *d*-Y6. ^1H NMR (400

486 MHz, Chloroform-*d*) δ 9.17 (s, 1H), 8.79 (s, 1H), 7.97 (s, 1H), 3.21 (s, 2H) ppm; ^{13}C
 NMR (151 MHz, Chloroform-*d*) δ 186.42, 158.89, 154.37, 147.68, 145.46, 139.71,
 488 139.31, 138.90, 137.93, 136.38, 136.21, 135.97, 134.04, 133.70, 131.01, 127.06,
 125.16, 120.08, 115.19, 114.71, 113.81, 68.90, 29.80, 29.47 ppm.

The GISANS data of PM6:*d*-Y7 was added to **Supplementary Fig. 23**.

**Supplementary Fig. 23** 2D TOF-GISANS patterns of **a** PM6:Y7 and **b** PM6:*d*-Y7 with
 their horizontal linecuts (dots) with best fits (solid line) shown in **c** and **d**, respectively.

We updated domain sizes in Supplementary Table 5.

**Supplementary Table 5** Morphology parameters fitted from the GISANS intensity
 profiles ($2R_{gc}$ is the crystallized acceptor domain size, $2R_{ga}$ is the amorphous acceptor
 domain size in the intermixing phase.)

Active layer	$2R_{gc}$ (nm)	$2R_{ga}$ (nm)
PM6: d -Y7	13.8	5.4
PM6: d -IDIC	22	N/A
P3HT: d -Y6	33.7	7.5

PTB7-Th: d -Y6	31.7	6.7
J71: d -Y6	28.7	6.7

We modified the following sentences on line 124, page 6 of the main text.

Similar aggregates were observed in the blend films of *d*-Y6 with other polymer donors
as well as in the blend film of PM6 with deuterated Y7 (*d*-Y7), a chlorinated Y6
derivative⁴⁷, yet no such scattering feature was observed in the film of PM6 blended
with another deuterated NFA – IDIC⁴⁸.

We modified the following sentences on line 355, page 20 of the main text.

To find out whether this short-range aggregation behavior is unique to Y-series NFAs,
we performed deuteration substitution on Y7, a chlorinated Y6 derivative⁴⁷, and IDIC,
another high-performance NFA modified from ITIC⁴⁸. The synthesis of *d*-Y7 follows
the same route as *d*-Y6 while the synthesis route of *d*-IDIC can be found in
**Supplementary Fig. 1d**. The corresponding ¹H and ¹³CNMR spectra are shown in
**Supplementary Fig. 5-8**, while the mass spectroscopy and elemental analysis for *d*-
IDIC can be found in **Supplementary Fig. 9** and notes of Supporting Information,
respectively. Upon deuteration, the neutron SLDs increase from 1.77×10^{-6} to 5.37×10^{-6}
\AA^{-2} and 1.85×10^{-6} to $1.09 \times 10^{-5} \text{\AA}^{-2}$ for Y7 and IDIC, respectively. However, the
scattering feature related to short-range aggregates was only detected in the blend film
of PM6:*d*-Y7 but not in PM6:*d*-IDIC, as shown in **Supplementary Fig. 23, 24** and
**Supplementary Table 5**.

We added the following sentences to line 393, page 23 of the main text.

Since this unique feature has been observed in blend films of both Y6 and Y7, it likely

arises from their excellent backbone planarity compared to ITIC-based NFAs. It has
been shown that the beta side chains attached to the outer core groups of Y6 molecules,
regardless of their chain lengths and shapes, can induce steric hindrance effects to
suppress dihedral angles between core and end groups^{61,62}. Therefore, we anticipate that
our conclusion should be generally applicable to Y-series NFAs with side chains
attached to the same position.

We added the following references to the main text.

Wu, J. *et al.* Exceptionally low charge trapping enables highly efficient organic
bulk heterojunction solar cells. *Energy & Environmental Science* **13**, 2422-2430
(2020).

Luke, J., Yang, E. J., Labanti, C., Park, S. Y. & Kim, J.-S. Key molecular
perspectives for high stability in organic photovoltaics. *Nature Reviews*
*Materials* **8**, 839-852 (2023).

We added the following co-authors who synthesized *d*-Y7 to the author list.

Guilong Cai^{1,3,10}, Yuhao Li^{1,2,10*}, Yuang Fu^{1,10}, Hua Yang², Le Mei⁴, Zhaoyang Nie⁵,

Tengfei Li⁶, Heng Liu¹, Yubin Ke², Xun-Li Wang^{7,8}, Jean-Luc Brédas⁹, Man-Chung

Tang⁵, Xiankai Chen⁴, Xiaowei Zhan^{6,*} and Xinhui Lu^{1,*}

⁵ Institute of Materials Research, Tsinghua Shenzhen International Graduate School,

Tsinghua University, 518055 Shenzhen, China.

We included their contributions to this manuscript on line 557, page 30 of the main text.

Z.N., M.-C.T. synthesized *d*-Y7.

7 . The ¹³C NMR and elemental analysis of new compounds need to be provided.

**Reply:** We thanks the reviewer for this important suggestion. We have added the ¹³C
NMR spectra and elemental analysis results of *d*-Y6 and *d*-IDIC to the Supporting
Information.

**Supplementary Fig. 3** ¹³C NMR spectrum of *d*-Y6.

**Supplementary Fig. 8** ^{13}C NMR spectrum of *d*-IDIC.

***d*-Y6.** To a three-necked round bottom flask were added Compound 5 (220 mg, 0.2

568 mmol), *d*-2FIC (140 mg, 0.6 mmol), pyridine (0.15 mL) and chloroform (25 mL). The

569 mixture was deoxygenated with nitrogen for 20 min and then stirred at reflux for 12 h.

After cooling to room temperature, the mixture was poured into methanol (200 mL)

and filtered. The residue was purified by column chromatography on silica gel using a

mixture solvent as eluent (petroleum ether/dichloromethane, v/v = 1/1) to give a blue

solid (255 mg, 83%). ^1H NMR (400 MHz, Chloroform-*d*) δ 9.2 (s, 2H), 3.2 (s, 4H). ^{13}C

NMR (100 MHz, Chloroform-*d*) δ 186.3, 159.0, 154.1, 147.7, 145.3, 137.9, 136.1,

135.4, 134.6, 134.0, 133.3, 130.7, 120.1, 115.1, 114.7, 113.7, 68.9, 29.8. HRMS

(MALDI) calculated for $\text{C}_{82}\text{H}_6\text{D}_{80}\text{F}_4\text{N}_8\text{O}_2\text{S}_5$: 1531.0429; found: 1531.0424 (M^+).

Elemental analysis calculated for $\text{C}_{82}\text{H}_7\text{D}_{80}\text{F}_4\text{N}_8\text{O}_2\text{S}_5$: C, 64.23; N, 7.31. Found: C,

64.12; N, 7.13.

***d*-IDIC.** To a three-necked round bottom flask were added Compound S12 (142 mg,
0.2 mmol), *d*-IC (120 mg, 0.6 mmol), pyridine (0.15 mL) and chloroform (25 mL). The
mixture was deoxygenated with nitrogen for 20 min and then stirred at reflux for 12 h.
After cooling to room temperature, the mixture was poured into methanol (200 mL)
and filtered. The residue was purified by column chromatography on silica gel using a
mixture solvent as eluent (petroleum ether/dichloromethane, v/v = 1/1) to give a blue
solid (173 mg, 81%). ¹H NMR (400 MHz, Chloroform-*d*) δ 9.0 (s, 2H), 7.7 (s, 2H), 7.6
(s, 2H). ¹³C NMR (100 MHz, Chloroform-*d*) δ 188.5, 160.9, 160.4, 157.7, 156.8, 141.3,
140.1, 138.7, 138.1, 137.9, 137.1, 122.2, 116.2, 115.0, 114.9, 69.1, 54.2. HRMS
(MALDI) calculated for C₆₆H₆D₆₀N₄O₂S₂: 1070.8388; found: 1070.8384 (M⁺).
Elemental analysis calculated for C₆₆H₆D₆₀N₄O₂S₂: C, 73.96; N, 5.23. Found: C, 73.56;
590 N, 5.07.

We modified the following sentence on line 145 of page 9.

The new compounds were fully characterized using ¹H and ¹³C nuclear magnetic
resonance (NMR) spectroscopy, mass spectrometry, and elemental analysis as shown
in Fig. 1b, Supplementary Fig. 2-4, and note in Supporting Information, respectively.

REVIEWERS' COMMENTS

Reviewer #1 (Remarks to the Author):

The authors have been fully addressed my concerns. The manuscript could be published as this version.

Reviewer #2 (Remarks to the Author):

The authors have carefully addressed all the reviewer comments and have substantially improved their literature survey.

In the discussion of their results the authors were now able to convince me about the originality of their work. In particular, is original the combination of NFA deuteration and GISANS to unravel the short-range aggregation of Y6 molecules embedded within the amorphous intermixing phase of polymer/Y6. As far as I know, this type of morphology has not been previously reported in BHJ OPVs. This work highlights the unique morphological advantage of Y6 and the important role of short-range aggregation in OPV performance.

I recommend this work to publish in Nature Communications in its current form.

Reviewer #3 (Remarks to the Author):

The authors addressed comments and amended the manuscript accordingly. Therefore I recommend its publication.

Reviewer #4 (Remarks to the Author):

In the revised manuscript, although part of the problems was addressed, some of key issues remain unaddressed clearly.

1) The use of deuterated materials is costly and does not improve efficiency, has little application and is too costly, please give a cost analysis for the synthesis of corresponding deuterated materials in comparison with the conventional materials.

2) d-L8-BO and d-BO-4Cl are more popular and more efficient Y-series acceptors and they showed different amount and type of deuterated atoms, the corresponding experiment for d-L8-BO and d-BO-4Cl will be more support their conclusion.

3) Carbon-fluorine coupling (in ^{13}C) for d-Y6 and d-L8-BO should be included in NMR data.

Responses to comments from reviewers

We would like to thank the reviewers and the editor for their valuable time and comments, which have greatly helped to improve the quality of our paper. We have addressed the comments from each reviewer point by point and revised the manuscript accordingly. Our detailed responses are shown in blue and the corresponding revisions in both the manuscript and supplementary information are indicated with yellow highlights.

Reviewer #1:

Comments:

The authors have been fully addressed my concerns. The manuscript could be published as this version.

Reply: We thank the reviewer for the positive comments.

Reviewer #2:

Comments:

The authors have carefully addressed all the reviewer comments and have substantially improved their literature survey.

In the discussion of their results the authors were now able to convince me about the originality of their work. In particular, is original the combination of NFA deuteration and GISANS to unravel the short-range aggregation of Y6 molecules embedded within the amorphous intermixing phase of polymer/Y6. As far as I know, this type of morphology has not been previously reported in BHJ OPVs. This work highlights the unique morphological advantage of Y6 and the important role of short-range aggregation in OPV performance.

I recommend this work to publish in Nature Communications in its current form.

Reply: We appreciate the reviewer's insightful comments and suggestions.

Reviewer #3:

Comments:

The authors addressed comments and amended the manuscript accordingly. Therefore I recommend its publication.

Reply: We really appreciate the reviewer's positive comments and insightful suggestions.

Reviewer #4:

Comments:

In the revised manuscript, although part of the problems was addressed, some of key issues remain unaddressed clearly.

1) The use of deuterated materials is costly and does not improve efficiency, has little application and is too costly, please give a cost analysis for the synthesis of corresponding deuterated materials in comparison with the conventional materials.

Reply: We agree with the reviewer that the deuteration of OPV active layer materials is costly. However, we meant to apply it as a labelling technique to probe the morphology of novel organic blend films so the material consumption per project would be low. We quote from the work by Li et al. [*Synth. Met.* 281, 116904 (2021).] that the synthesis cost of Y6 molecules is 1000 USD/g. Since the prices of deuterated components are on average 2 to 3 times higher than the corresponding non-deuterated ones, we estimate that the cost of *d*-Y6 molecules would be around 2500 USD/g. In this project, we have only used around 50 mg of *d*-Y6, corresponding to a total cost of \$125. Therefore, we believe that targeted deuteration is cost-worthy and could be applied to study other novel organic optoelectronic materials in the future.

2) d-L8-BO and d-BO-4Cl are more popular and more efficient Y-series acceptors and they showed different amount and type of deuterated atoms, the corresponding

experiment for d-L8-BO and d-BO-4Cl will be more support their conclusion.

Reply: We thank the reviewer for raising this important concern. We chose to deuterate Y7, because it has similar chemical structure to Y6, as an easy and fast showcase for the generality of our work. In the coming work, we indeed to plan to extend our study to more high-performance Y-series molecules such as L8-BO and BO-4Cl to establish structure-performance relationships.

3) Carbon-fluorine coupling (in ^{13}C) for d-Y6 and d-L8-BO should be included in NMR data.

Reply: We thank the reviewer for this important comment. We have added the enlarged ^{13}C NMR spectrum of *d*-Y6 (between 150 and 160 ppm) to the inset of **Supplementary Fig.3**. The presence of C-F coupling is evidenced by the doublet at δ 155.64 (d, $J=16.2$ Hz), 153 (dd, $J=14.8, 7.3$ Hz).

Supplementary Fig.3 has been modified as shown below.

Supplementary Fig. 3 ^{13}C NMR spectrum of *d*-Y6. The inset is the enlarged spectrum

between 152 and 156 ppm to highlight the presence of C-F coupling.

We have also added the following comments in Supporting Information.

^{13}C NMR (100 MHz, Chloroform-*d*) δ 186.3, 159.0, 154.1, 147.7, 145.3, 137.9, 136.1, 135.4, 134.6, 134.0, 133.3, 130.7, 120.1, 115.1, 114.7, 113.7, 68.9, 29.8. The presence of C-F coupling is evidenced by the doublet at δ 155.64 (d, $J=16.2$ Hz), 153 (dd, $J=14.8$, 7.3 Hz).